# Resonant cavity phosphor

Tae-Yun Lee ®[1,2], Yeonsang Park ®[3,4] ✉ & Heonsu Jeon ®[1,2,5] ✉

While *phosphors* play an immensely important role in solid-state lighting and full-colour displays, it has been noted lately that their performance can be largely improved via structural engineering. Here, phosphor material is synergistically merged with yet another structurally engineered platform, *resonant cavity* (RC). When a 40-nm-thick colloidal quantum dot (CQD) film is embedded in a tailored RC with a moderate cavity quality factor ($Q \approx 90$), it gains the ability to absorb the majority (~87%) of excitation photons, resulting in significantly enhanced CQD fluorescence (~29×) across a reasonably broad linewidth (~13 nm). The colour gamut covered by red and green pixels implemented using the RC phosphor—along with a broad bandwidth (~20 nm) blue excitation source—exceeds that of the sRGB standard (~121%). The simple planar geometry facilitates design and implementation of the RC phosphor, making it promising for use in real applications.

Phosphors, which can convert the colour of photons through sequential absorption and emission, have been actively researched over the years[1]. In particular, the emergence of phosphor-capped white light-emitting diodes (LEDs)[2,3] has enabled and boosted important application sectors, such as solid-state lighting and modern full-colour displays, and thus reinforced the demand for phosphors. The overall performance of a phosphor can be represented by its *external quantum efficiency* (EQE), expressed as $\eta_{ext} = A \times \eta_{int} \times \eta_{oc}$, where $A$, $\eta_{int}$, and $\eta_{oc}$ are the absorbance (of excitation photons by the phosphor), internal quantum efficiency (in converting the absorbed excitation photons to emissive photons of different wavelengths inside the phosphor), and out-coupling efficiency (in extracting the emitted photons from the phosphor), respectively. Thus far, phosphors have been developed primarily excavating new materials with higher $\eta_{int}$, which culminated in the synthesis[4-6] and adoption[7] of colloidal quantum dots (CQDs). Currently, progress in the material-oriented phosphor development process has been limited. As a bypass to the stalemate, attention to the structural aspects of phosphors was given to improve the phosphor performance via $A$ or $\eta_{oc}$. Examples include incorporating high-index scattering centres, such as nano-rods[8], a mesoporous film[9], or metal-oxide nanoparticles[10-12], and adding a dielectric distributed Bragg reflector (DBR) with its centre wavelength tuned to either excitation[13,14] or down-converted emission[14]. However, the

bottom-up approaches in the former are too complex to analyse or design systematically, whereas the top-down approaches in the latter do not have scope for further development.

We previously proposed and demonstrated the concept of *structurally engineered phosphors*[15-18], in which the phosphor material was carved into a nanophotonic structure to significantly increase $A$ (while $\eta_{int}$ and $\eta_{oc}$ remained unaltered). More specifically, we incorporated CQDs, which served as the phosphor material, into a prefabricated planar photonic crystal (PhC) backbone such that the Γ-point ($k_{\parallel} = 0$) band-edge mode(s) associated with the resultant PhC structure was tuned at the wavelength for CQD excitation. The *PhC phosphor* thus prepared exhibited greatly enhanced CQD fluorescence because of the resonant absorption of the vertically incident excitation photons. Notably, the concept of structural engineering (for enhanced $A$) is not bound to any specific phosphor material and is thus compatible with the traditional efforts of developing new phosphor materials (for higher $\eta_{int}$).

In this Article, we propose simple one-dimensional (1D) *resonant cavity* (RC), which comprises a central cavity layer and two DBR mirrors on both sides, as yet another platform for structurally engineered phosphors. The RC structure, although old-fashioned, has recently been revaluated as a topological photonic system with functional robustness[19]. Cavity quantum electrodynamics suggests that the optical transition rates of a quantum system embedded in

[1]Department of Physics and Astronomy, Seoul National University, Seoul 08826, Republic of Korea. [2]Inter-university Semiconductor Research Centre, Seoul National University, Seoul 08826, Republic of Korea. [3]Department of Physics, Chungnam National University, Daejeon 34134, Republic of Korea. [4]Institute of Quantum Systems, Chungnam National University, Daejeon 34134, Republic of Korea. [5]Institute of Applied Physics, Seoul National University, Seoul 08826, Republic of Korea. ✉e-mail: yeonsang.park@cnu.ac.kr; hsjeon@snu.ac.kr

an RC can be significantly altered. The spontaneous emission rate, for example, is known to be modified by the available photonic density of states (PDOS) in the environment[20], implying that radiation from an electric dipole inside an RC is either enhanced or suppressed depending on the fulfilment of resonant conditions[21]; RC-LEDs with improved electroluminescence properties are the most representative photonic devices of this type[22]. Similar effects are expected for the induced optical transition (stimulated emission or absorption) as well because the associated optical transition rate is still proportional to the PDOS[23], albeit it also depends on the number of existing radiation quanta (photons). Consequently, the induced transition rate can be significantly enhanced by placing dipoles at the antinode(s) of an RC mode where the PDOS is at its maximum. Relevant photonic devices include vertical-cavity surface-emitting lasers (VCSELs)[24], RC photodetectors, and RC photovoltaic cells[25–28]. Nonetheless, RC has never been explored or considered as a platform for phosphors. The *RC phosphor* (RC-PSP) proposed herein differs from the PhC phosphor previously developed by us[15–18]; it utilises a localised vertical cavity mode whereas the PhC phosphor relies on a periodic and extended in-plane band-edge mode. Furthermore, the RC-PSP is superior to the PhC phosphor in terms of both structure and performance. The RC-PSP can be constructed by vertically stacking planar layers one after another, which is far easier than the sophisticated lateral submicron-patterning process required for fabricating the PhC phosphor. As for the performance, the absorption of excitation photons is greater in the RC-PSP than in the PhC phosphor as revealed hereafter.

## Results

### Structural aspects of RC phosphor

Figure 1a shows the schematic of a typical colour pixel configuration used in modern full-colour display[29], in which a phosphor layer absorbs shorter-wavelength excitation photons from a micro-LED and converts them into longer-wavelength photons. For phosphor materials, chemically synthesised monodisperse CQDs are the most preferred candidates owing to their various advantages, including convenient colour tuning, high quantum efficiency, broad absorption bands, narrow emission linewidths, and fast modulation

response[30]. CQDs are typically dispersed in a polymer (30–40 wt%), which is then cast into a thick (~10 μm) phosphor film[31]. Despite the large amount of CQDs involved, a substantial fraction of excitation photons passes through the CQD phosphor layer without being absorbed, deteriorating the colour purity that CQD fluorescence could otherwise offer. To eliminate the 'blue' leakage, a pigment-based absorbing colour filter with a thickness of approximately 1 μm is often inserted[29,32], as shown in Fig. 1a. However, the additional blue filter layer makes the overall pixel configuration bulkier and complicates the fabrication. In this context, a desirable form factor for phosphors should be a thin film containing costly CQDs as few as possible yet capable of absorbing most excitation photons, which would obviate the need for an extra colour filter to absorb blue. We claim that embedding a thin CQD film inside the RC can help accomplish this seemingly impossible task.

Our RC-PSP has an asymmetric 1D RC structure, as shown in Fig. 1b, in which an $\lambda/2$-thick cavity layer is sandwiched between two dielectric DBRs: the emission-side DBR (emDBR; adjacent to the substrate) and the excitation-side DBR (exDBR; adjacent to the air). The asymmetry in the cavity was intentionally incorporated to facilitate efficient excitation through exDBR. The overall RC-PSP structure was designed such that cavity resonance occurred at $\lambda_0 = 450$ nm, assuming a blue LED or laser diode (LD) as a convenient and compact phosphor excitation source. The cavity layer comprises a CdSe-based dense CQD film ($d_{CQD} \approx 40$ nm, $n_{CQD} \approx 1.82$) and two SiO$_2$ wing layers ($d_w \approx 40$ nm) on both sides. The two wing layers save costly CQD materials near the nodal planes in the RC mode where the electric field strength is weak. The two DBRs consist of alternating $\lambda/4$-thick dielectric layers of TiO$_2$ ($d_{TiO} \approx 49$ nm, $n_{TiO} \approx 2.32$) and SiO$_2$ ($d_{SiO} \approx 77$ nm, $n_{SiO} \approx 1.47$), starting and ending with the TiO$_2$ layers for the higher index contrasts to the environmental materials, air and silica. It should be noted that, compared to the conventional phosphor (Con-PSP) structure shown in Fig. 1a, the RC-PSP structure is only ~1/10 of its physical thickness and contains only ~1/100 of the net CQD amount[31].

### Enhanced absorption of excitation photons by RC phosphor

The transmittance (T), reflectance (R), and absorbance (A) spectra of the RC-PSP were calculated for various combinations of layer

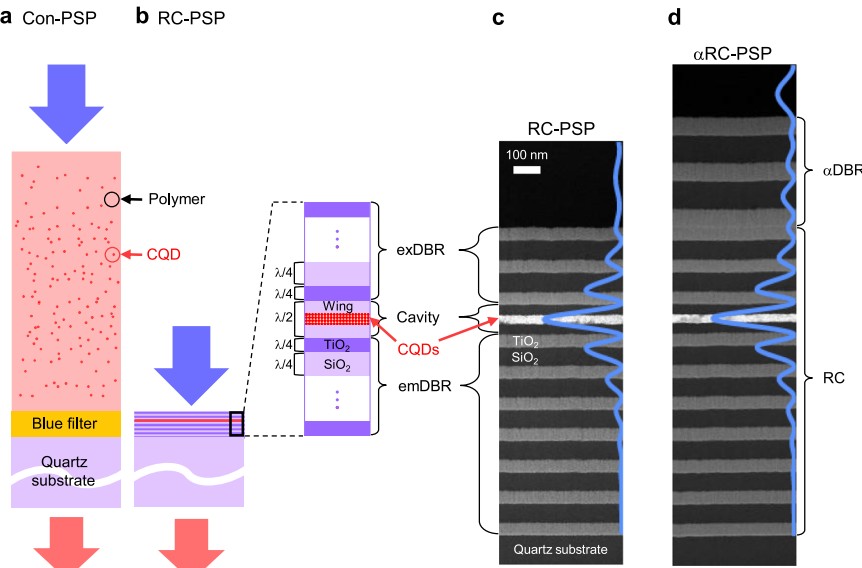

**Fig. 1 | Structure of the RC-PSP. a, b** Schematics of the colour pixel configurations based on a thick-film conventional phosphor (Con-PSP; with no RC structure) (**a**) and the resonant cavity phosphor (RC-PSP) (**b**). The inset in **b** illustrates structural details of the RC-PSP. **c, d** Cross-sectional TEM images of the RC-PSP (**c**) and αRC-PSP (**d**). Calculated modal profiles set in the corresponding RC structures are drawn over the images. The prefix 'α' stands for 'augmenting' or 'augmented'.

numbers in the two DBRs (Supplementary Information S1). The extinction coefficient of the CQD film determined from independent spectroscopic ellipsometry measurements was used in the calculations (Supplementary Information S2). The peak absorbance and resonance linewidth for the RC-PSP were closely coupled and dependent on DBR layer numbers. Therefore, the appropriate RC structure that best fits given excitation requirements−for example, by an LED or LD−must be determined. This study chose the DBR layer numbers as $N_{ex}$ = 5 (or 2.5 pairs) for exDBR and $N_{em}$ = 13 (or 6.5 pairs) for emDBR because the resulting RC-PSP structure offers high values for all three different absorption parameters−the peak, integrated, and weighted absorbances−with a relatively small number of DBR layers $N_{ex} + N_{em}$ (Supplementary Information S3). As shown in Fig. 2a(ii), the RC-PSP exhibits a peak absorbance $A_{max} \approx 0.87$ and a resonance linewidth $\Delta\lambda_0 \approx 7$ nm in the full-width at half-maximum (FWHM); the corresponding *cold* cavity, having the same RC structure but no absorbing ability, results in a quality factor $Q \approx 90$. Conversely, the reference phosphor (Ref-PSP), which corresponds to the cavity layer itself−a 40-nm-thick CQD layer cladded by the two $SiO_2$ wing layers on both sides, exhibits a fairly flat spectral profile with no resonance feature, and the absorbance

at $\lambda_0 = 450$ nm is only $A \approx 0.03$−Fig. 2a(i). The absorption enhancement factor (AEF), defined as the absorbance ratio between RC-PSP and Ref-PSP, was deduced as a function of wavelength, as shown in Fig. 2d. The resonantly enhanced absorption, which is a hallmark feature of RC-PSP, was as large as $AEF_{max} \approx 32$ at resonance.

## Experimental implementation of RC phosphor

The fabrication of RC-PSP is similar to that of an optically excited CQD VCSEL[33]. First, the emDBR and one of the wing layers were sequentially vacuum-deposited on a fused quartz substrate. The CdSe-ZnS core-shell CQDs emitting at $\lambda_R \approx 620$ nm were, after dispersed in hexane, spin-coated to form a 40-nm-thick film on top of the already deposited wing layer. Another round of sequential vacuum deposition for the other wing layer and exDBR completed device fabrication. Figure 1c shows a cross-sectional transmission electron microscopy (TEM) image of the RC-PSP fabricated as described. For comparison, Ref-PSP that lacked of the both DBRs was prepared on a separate fused silica substrate. T and R spectra were measured for both, while A was deduced from the measured T and R using the relationship $A = 1 - T - R$ (Supplementary Information S4). All the experimental results agree well with the calculated

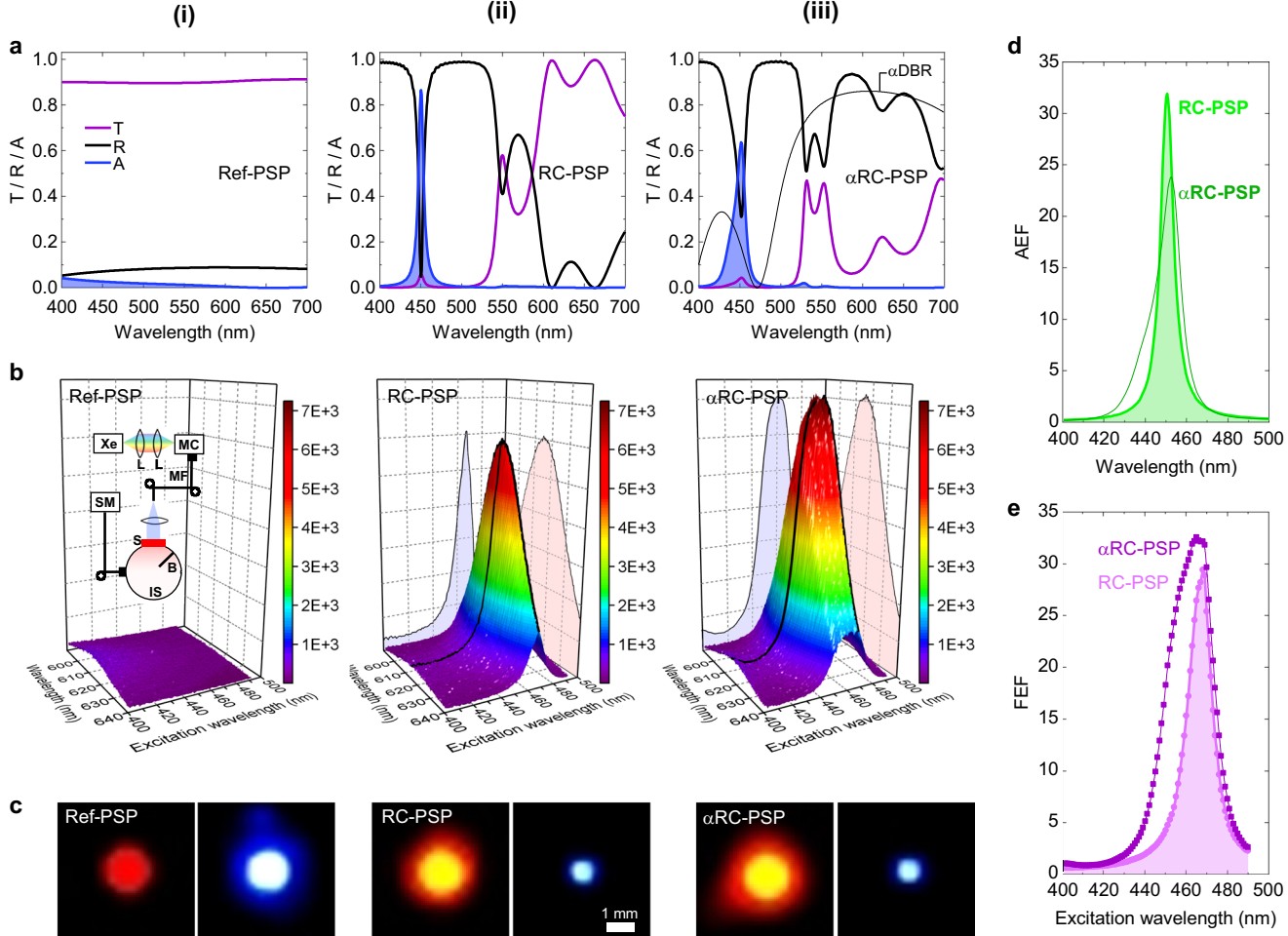

**Fig. 2 | Characteristics of the RC-PSP and αRC-PSP. a** Calculated transmittance (T), reflectance (R), and absorbance (A) of (i) the reference phosphor (Ref-PSP), (ii) RC-PSP, and (iii) αRC-PSP. The reflectance spectrum of the αDBR for the αRC-PSP is also shown in (iii). **b** Photoluminescence excitation (PLE) data for (i) the Ref-PSP, (ii) RC-PSP, and (iii) αRC-PSP, all measured with excitation linewidth $\delta\lambda_{ex}$ = 2 nm. Shown as inset in (i) is the schematic of the PLE measurement setup. (Xe: xenon lamp, MC: monochromator, IS: integrating sphere, SM: spectrometer, L: lens, MF: multimode fiber, B: baffle, S: sample). **c** Colour photographs captured from the top

while excited resonantly ($\lambda_{ex} = \lambda_0 \approx 468$ nm) for (i) the Ref-PSP, (ii) RC-PSP, and (iii) αRC-PSP. In each, the left panel was taken with a low-pass filter (LPF) ( > 575 nm) inserted, whereas the right panel was taken with a short-pass filter (SPF) ( < 550 nm). **d** Absorption enhancement factor (AEF) spectra for the RC-PSP and αRC-PSP, deduced from the calculated absorbance spectra in **a**. **e** Fluorescence enhancement factor (FEF) spectra for the RC-PSP and αRC-PSP, deduced from the experimental PLE data in **b**.

ones in Fig. 2a(i) and 2a(ii), except that the RC resonance is spectrally shifted to $\lambda_0 \approx 468$ nm and the linewidth is broadened to $\Delta\lambda_0 \approx 13$ nm due to *inhomogeneous broadening*.

## Photoluminescence excitation measurements

Photoluminescence excitation (PLE) measurements were performed to characterise RC-PSP. The inset of Fig. 2b(i) schematically depicts the PLE setup, in which a custom-built tunable excitation source based on a Xe lamp was combined with an integrating sphere, which was employed to capture all the photons emanating from the emission-side hemisphere through the fused silica substrate. The PLE data measured with excitation linewidth $\delta_{ex} = 2$ nm are plotted in Fig. 2b(i) and 2b(ii) for Ref-PSP and RC-PSP, respectively. As shown in Fig. 2b(ii), the CQD fluorescence from the RC-PSP is resonantly enhanced at $\lambda_{ex} \approx 468$ nm. The resonant excitation wavelength matched perfectly with the cavity resonance wavelength $\lambda_0$ identified from the independently measured T and R spectra, indicating that the enhanced CQD fluorescence was a direct consequence of the resonant absorption of the excitation photons by the RC. In contrast, Ref-PSP exhibited neither resonance nor enhancement of CQD fluorescence—Fig. 2b(i). The fluorescence enhancement factor (FEF) is shown as a function of excitation wavelength in Fig. 2e. It is defined as the CQD fluorescence intensity ratio between the RC-PSP and Ref-PSP, and can be deduced experimentally from Fig. 2b(i) and 2b(ii). Note that FEF is roughly equivalent to the EQE ratio, $\eta_{ext}^{RC} / \eta_{ext}^{Ref}$, as $\eta_{int}$ and $\eta_{oc}$ are not altered by the presence of RC. The FEF reached its maximum value $FEF_{max} \approx 29$ at $\lambda_{ex} = \lambda_0 \approx 468$ nm.

## Performance-augmented RC phosphor

The performance of the RC-PSP can be further improved by a modified RC structure. As a proof of concept, an *augmenting* DBR (αDBR) composed of 2.5 pairs of $TiO_2$ ($d_{TiO} \approx 68$ nm) and $SiO_2$ ($d_{SiO} \approx 104$ nm) layers was added on top of the exDBR to reflect the CQD fluorescence emanating from the excitation side back to the emission side[13,14]. The calculated T, R, and A spectra of the resultant *augmented* RC-PSP (αRC-PSP), along with the calculated reflectance spectrum of the αDBR, which exhibits $R \approx 0.8$ at $\lambda_R \approx 620$ nm, are shown in Fig. 2a(iii). Note that the overall resonance features near $\lambda_0 \approx 450$ nm are unchanged—albeit $A_{max}$ is lowered and $\Delta\lambda_0$ is broadened due to partial reduction in the cavity quality by the addition of the αDBR. However, linewidth broadening is not necessarily a disadvantage because phosphors are typically excited by a light source with a broad bandwidth, such as an LED, for which an RC with a comparable resonance linewidth is preferable to maximise the use of excitation photons. The AEF of the αRC-PSP deduced from Fig. 2a(i) and 2a(iii) is also presented in Fig. 2d.

The αRC-PSP was also experimentally implemented. Its cross-sectional TEM image is shown in Fig. 1d. The T, R, and A spectra measured for the αRC-PSP (Supplementary Information S4) are in excellent agreement with the calculated spectra in Fig. 2a(iii). PLE measurements were repeated for the αRC-PSP; the results are plotted in Fig. 2b(iii). Again, the CQD fluorescence intensity was resonantly enhanced. The FEF spectrum of αRC-PSP, deduced from Fig. 2b(i) and 2b(iii), is shown in Fig. 2e. Despite the lower maximum absorbance in Fig. 2d, the αRC-PSP exhibits better fluorescence performance than the RC-PSP in terms of both maximum FEF ($FEF_{max} \approx 33$) and linewidth ($\Delta\lambda_0 \approx 26$ nm) due to the presence of the αDBR. Further ingenious structural modifications to the overall RC could improve the performance.

## Visual demonstration

For direct visual demonstrations, three types of CQD phosphors—the Ref-PSP, RC-PSP, and αRC-PSP—were photographed under excitation at various wavelengths (Supplementary Information S5). Figure 2c shows selected photographs under resonant excitation

**Table 1 | Peak and integrated fluorescence intensities for Ref-PSP, RC-PSP, and αRC-PSP**

|  | Ref-PSP | RC-PSP | αRC-PSP |
|---|---|---|---|
| Peak Intensity | 190 | 6,070 | 7,220 |
| Integrated Intensity | 6530 | 191,650 | 209,720 |

conditions at $\lambda_{ex} = \lambda_0 \approx 468$ nm. The left panel for each PSP was captured with a long-pass filter (LPF; $\lambda > 575$ nm) inserted in front of the camera to capture only the red CQD fluorescence. The CQD fluorescence intensities of RC-PSP and αRC-PSP significantly exceeded that of Ref-PSP. For quantitative comparisons, both the peak and integrated fluorescence intensities for Ref-PSP, RC-PSP, and αRC-PSP are extracted from the PLE data in Fig. 2b and summarized in Table 1. The photographs in the right panels were captured with a short-pass filter (SPF; $\lambda < 550$ nm) so that only the blue leakage was visible. While Ref-PSP exhibited an intense blue leakage (due to low A and high T), RC-PSP and αRC-PSP only showed a dim blue leakage (due to high A and low T). All these visual observations completely agree with the T and A spectra in Fig. 2a and the PLE data in Fig. 2b.

## Red and green RC phosphors for display application

Although efficient white light generation is certainly a possibility, the most obvious and impactful application sector for RC-PSP should be the development of full-colour displays based on hybrid micro-LED arrays, in which independently addressable red-green-blue (RGB) pixels can be prepared by combining blue LEDs with red and green phosphors[29], as depicted in Fig. 3a. It should be noted that the RC-PSP structures for the red and green pixels are nominally identical because their excitation sources and thus their excitation wavelengths are identical. However, to realise both red and green pixels from a common RC-PSP platform, the index contrast between the constituent DBR layers must be reduced. Otherwise, the stopband of the emDBR may extend to the green spectral region, making it difficult for the green CQD fluorescence to escape from the RC-PSP structure. Hence, the high-index DBR material $TiO_2$ was replaced with $Ta_2O_5$ ($n_{TaO} \approx 2.08$), while the numbers of DBR layers were increased to $N_{ex} = 7$ (3.5 pairs) and $N_{em} = 31$ (15.5 pairs) to compensate for the lowered index contrast. A batch of green CQDs with a peak fluorescence wavelength $\lambda_G \approx 530$ nm was prepared in addition to the red CQDs used in the concept demonstration.

In Fig. 3b and c, the fluorescence spectra measured directly from the red and green CQDs are compared with the measured reflectance spectrum of the $Ta_2O_5/SiO_2$ emDBR. As intended, the red and green CQD fluorescence peaks were outside the emDBR stopband, ensuring high transmittance of the red and green CQD fluorescence through the emDBR. PLE measurements were performed on the red and green RC-PSPs consisting of $Ta_2O_5/SiO_2$ DBRs (Supplementary Information S6). Unlike the previous PLE measurements, the excitation linewidth was intentionally broadened to $\delta\lambda_{ex} \approx 20$ nm to mimic an LED-like realistic excitation source[34,35]. As shown in Fig. 3b and c, the FEFs are as high as ~6.8 and ~5.9 for the red and green RC-PSPs, respectively. Compared with the results shown in Fig. 2e, which were obtained with $\delta\lambda_{ex} \approx 2$ nm, the maximum FEFs are much lower, while the resonance linewidths become larger ($\Delta\lambda_0 \approx 20$ nm).

## Colour rendering

Figure 4a shows the CIE 1931 colour space chromaticity diagram, in which the RGB colour coordinates rendered by the red and green RC-PSPs (under resonant conditions) and the blue excitation source were identified. The resulting colour triangle defines the colour gamut covered by our RGB sources. For comparison, two standard colour spaces used in the display industry, sRGB and NTSC, are also

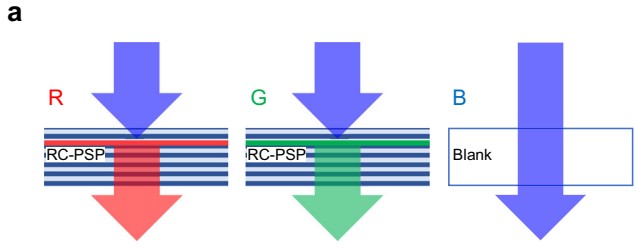

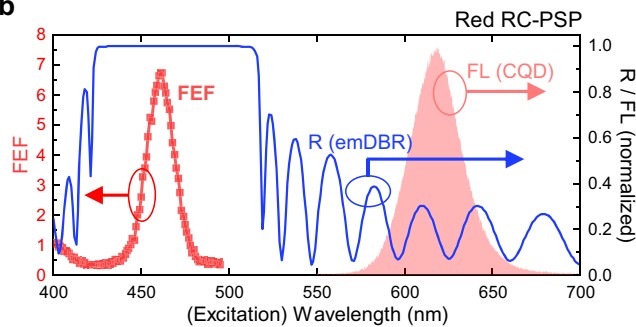

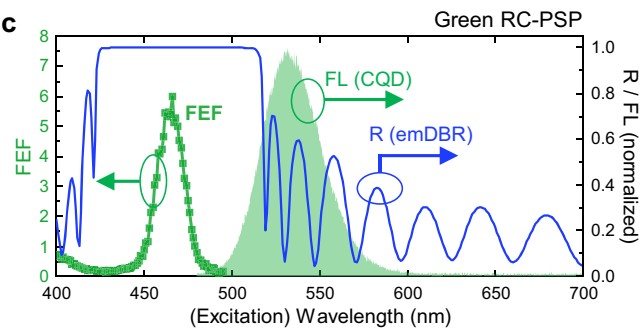

**Fig. 3 | RC-PSPs composed of Ta$_2$O$_5$/SiO$_2$ DBRs. a** Schematic of the RC-PSP based RGB colour pixels for full-colour displays. The combinations of the blue excitation sources and the RC-PSPs constitute the red and green pixels, while the blue excitation source itself serves as the blue pixel. **b, c** Experimentally determined FEF spectrum of the red RC-PSP (**b**) and green RC-PSP (**c**). The reflectance spectrum of the emDBR that constitutes the RC and the intrinsic CQD fluorescence spectrum (with no RC) are also shown in each figure. The excitation linewidth during the PLE measurements (and subsequent FEF determinations) was set at $\delta\lambda_{ex} = 20$ nm.

shown. Regarding the colour space area, our colour triangle covered ~121% of the sRGB and ~86% of the NTSC. The emission spectra of the corresponding RGB chromatic sources, which are shown in Fig. 4b–d, are dominated by distinct red, green, and blue emission peaks. The inset in each figure shows a photograph captured through an open output port on the integrating sphere, which vividly displays the colour purity rendered by each source. Nonetheless, Fig. 4b and c display small but finite blue peaks resulting from unabsorbed residual excitation photons, estimated to be approximately 4% and 13% of the red and green peaks, respectively, in terms of integrated intensity. Further reduction or complete elimination of the blue leakage is a subject for future study. In contrast, the Ref-PSPs containing identical amounts of CQDs allowed the majority (~90%) of blue excitation photons to pass through, resulting in the red and green pixels completely overwhelmed by the blue hue, as shown in Fig. 4e–g (and the inset photographs). The resultant chromaticity triangle, plotted in the inset of Fig. 4a, is unrecognisably small. It should be emphasised that all the spectra and photographs shown in Fig. 4b–g were captured without any type of colour filter.

## Discussion

We investigated a simple 1D RC as yet another platform for structurally engineered phosphors. The RC was designed such that its resonance wavelength could be tuned for the excitation of—rather than for the emission from—the phosphor material inserted in the centre of the cavity. We demonstrated that owing to the significantly enhanced light-matter interaction at the antinode of the RC mode, a 40-nm-thick CQD phosphor layer was sufficient to absorb most of the excitation photons, resulting in a CQD fluorescence enhancement of ~29 times compared with the reference without RC. We also demonstrated that further improvements are possible by ingeniously modifying the RC structure. To assess the applicability of the RC-PSP to full-colour displays, we experimentally improvised and characterised RGB pixels based on the RC-PSP. The resultant colour gamut attained with only ~1/100 of the CQD amount used in Con-PSP-based full-colour display pixels was comparable to, and therefore competitive with, industrial standards. The RC-PSP proposed and demonstrated in this study has a simple planar geometry that enables precise design for tailored performance and facilitates fabrication. Therefore, the RC-PSP platform not only offers superior performance but is also readily applicable owing to its simplicity in structure.

## Methods

### Calculations of transmittance, reflectance, and absorbance spectra

Although analytical techniques exist for dealing with RC with complex active layers[36], numerical simulations based on the finite-difference time-domain (FDTD) method were performed to imitate real experiments as closely as possible, using commercial software (Ansys Lumerical FDTD, ANSYS, Inc.). The complex refractive indices for the dielectric layers (SiO$_2$, TiO$_2$, and Ta$_2$O$_5$) and red/green CQD layers were determined from independent ellipsometry measurements and used in the simulations, while the refractive index of the fused quartz substrate was obtained from a handbook[37]. Because of the simple planar geometry of the RC-PSP, two-dimensional (2D) simulations were deemed sufficient. Plane-wave sources were used to mimic real excitation conditions, where the excitation photons were incident in the direction perpendicular to the phosphor plane. Planar monitors for transmittance and reflectance were placed after the RC (beyond the emDBR) and before the RC (in front of the exDBR), respectively, and a 2D box monitor encircling the CQD layer was used for absorbance.

### Device fabrication

Device fabrication involved three simple steps: two vacuum depositions and one spin-coating in between. First, the TiO$_2$/SiO$_2$ (or Ta$_2$O$_5$/SiO$_2$) emDBR and the first SiO$_2$ wing layer were deposited on a fused quartz substrate using an e-gun evaporator at an elevated temperature of T = 150 °C. During the deposition of the high-index layers, an O$_2$ environment was provided to fully oxidise the metal ions. Subsequently, commercially procured CdSe-ZnS core-shell CQDs (CZO-620H and CZO-530H, Zeus) were dispersed in a cyclohexane solution and spin-coated on top of the first SiO$_2$ wing layer. The CQD concentration (2 wt%) and spin speed (3000 rpm) were carefully selected to obtain the desired CQD film thickness (40 nm). Finally, another round of vacuum deposition of the second SiO$_2$ wing layer and exDBR was performed to complete the device fabrication. The first and last layers of both emDBR and exDBR was a high-index layer (TiO$_2$ or Ta$_2$O$_5$). For αRC-PSP, exDBR and αDBR were simultaneously deposited in a single run.

### Measurements of transmittance and reflectance spectra

A halogen illuminator system with a fibre-based light guide (FOK-100W, Fiber Optic Korea) was employed as a convenient white-light source for the entire visible spectral range. The diverging beam from the output tip of the fibre was then transformed into a parallel

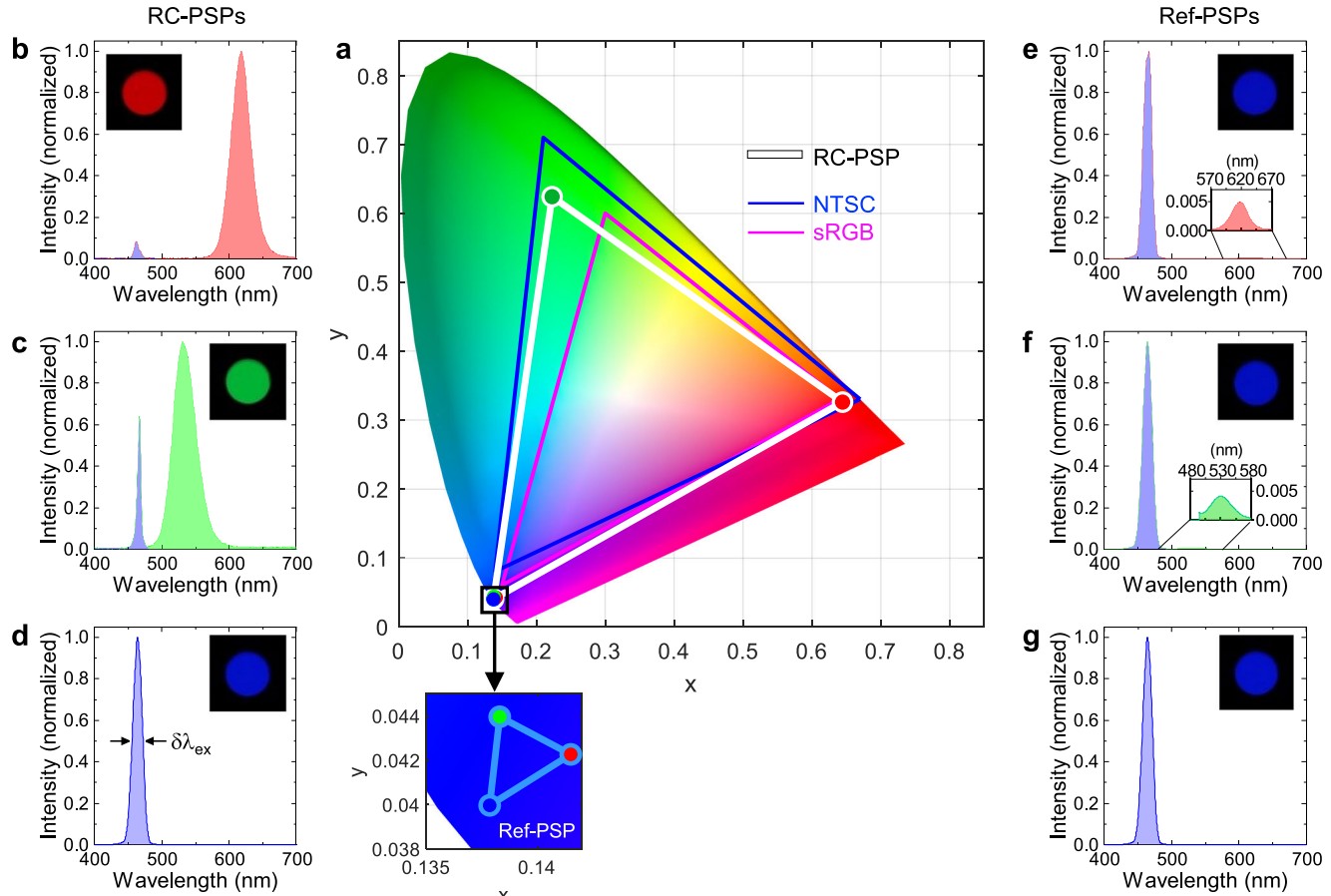

**Fig. 4 | Colour rendering by the RC-PSPs. a** Colour coordinates (and the corresponding colour triangles) rendered by the resonantly excited red and green RC-PSPs and the blue excitation source, plotted on the CIE 1931 colour space chromaticity diagram. Two colour triangles of the industrial standards, sRGB and NTSC, are also plotted for comparison. The colour triangle rendered by the Ref-PSPs, which is tightly localised in the blue corner, is magnified in the inset. **b–d** Total emission spectra captured by the integrating sphere for the red RC-PSP (**b**), green RC-PSP (**c**), and blue excitation source (**d**). **e–g** Total emission spectra captured by the integrating sphere for the red Ref-PSP (**e**), green Ref-PSP (**f**), and blue excitation source (**g**). Photographs of an open output port on the integrating sphere, taken for the corresponding pixel configurations, are shown as insets in **b–g**.

beam with a diameter of ~1 mm using a combination of a collimating lens and iris diaphragm. To measure both transmittance and reflectance at an incidence angle of 0°, a non-polarising beam-splitting cube with a 50:50 ratio was inserted in front of the sample. The transmitting and reflecting beams were coupled independently to the multimode fibres and sequentially fed into a spectrometer (HR4000CG-UV-NIR, Ocean Optics). The recorded spectra were then normalised using the reference spectra obtained either without the sample for transmittance or with a high-reflection (R > 0.99) broadband ($\lambda$ = 400–700 nm) dielectric mirror for reflectance.

**Photoluminescence excitation measurements**

An excitation source with a wide wavelength-tuning range was constructed by combining a Xe lamp (6271 Xenon Arc Lamp, Newport) and a monochromator (CM110, Spectral Products). The excitation linewidth $\delta\lambda_{ex}$ was adjusted over an FWHM range of 2–20 nm using the monochromator slit width. The excitation beam was guided to the sample located at the input port of the integrating sphere using a multimode fibre. The CQD fluorescence spectra from the sample were measured in transmission geometry while the excitation wavelength was scanned. During the measurements, the excitation beam was incident on the exDBR whereas the CQD fluorescence emanating from the other side (through the emDBR and fused quartz substrate) was collected using an integrating sphere (819D-SF-4, Newport). Care was taken to ensure that the sample was at the same level as the input port of the integrating sphere so that only the CQD fluorescence emitted

across the upper hemisphere could be captured, whereas the CQD fluorescence emitted across the lower hemisphere and guided through the silica substrate was excluded. The CQD fluorescence collected by the integrating sphere was fed into a spectrometer (Kymera 193i-A with an iVac 316 CCD, ANDOR) using a multimode fibre or photographed through an open output port.

## Data availability

The source data associated with the manuscript and Supplementary Information is provided with this paper, integrated into an Excel file with one subfigure per sheet. Source data are provided with this paper.

## Code availability

Some figures were generated using MATLAB code. The MATLAB codes are provided with this paper.

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

## Acknowledgements

This study was supported by the National Research Foundation of Korea (NRF-2020R1A2C2008583). Y.P. also acknowledges the support by the National Research Foundation of Korea (NRF-2020R1A6A1A03047771).

## Author contributions

T.-Y.L. conducted most of the experimental work, including design, fabrication, measurements, and simulations. Y.P. supervised the device fabrication and characterisation. H.J. conceived and directed the study. All the authors contributed to the scientific discussion and preparation of the manuscript.

## Competing interests

The authors declare no competing interests.
