## [Peer Review File · Nature Communications]

Reviewers' Comments:

Reviewer #1:

Remarks to the Author:

The work titled "Resonant cavity phosphor" by TAE-YUN LEE et al.. In this work one-dimensional resonant cavity that comprises a central cavity layer and two DBR mirrors on both sides as a platform for structurally engineered phosphors was fabricated and characterized. Three types of CQD phosphors were fabricated one without Bragg reflector and the other two phosphors fabricated with varying the thicknesses of Bragg layers. Overall, the work may be of interest to a broader audience. However, the authors should address the following points outlined below to improve the scientific quality. After carefully addressing the suggested revisions, this work may be considered for publication in the respected, Nature Communications Journal.

- 1- The authors should add details about the bare CQD synthesis procedure or add references that include the synthesis information.
- 2- The fluorescence spectra of the quantum dots should be added to the text or Supplementary Information.
- 3- The symbol DRB line 120, page 6, should be corrected.
- 4- The results of absorption calculation showed that the absorption maximum located at 450 nm, and the excitation maximum located at 468 nm, whereas the emission located in the red region with high Stokes shift. These results indicate that the emission from quantum dots originating from trapping state and this contradict with the core-shell structure, could the authors explain?

Reviewer #2:

Remarks to the Author:

In this work Jeon et. al. proposes a one dimensional resonant cavity, including a cavity layer and two DBR mirrors, for structurally engineered phosphor demonstration.

The work presented is a simple yet an interesting approach to tailor the overall light output for pixelated color converters with a Q factor of around 90. The results are believed to open up new applications and designs for the pixelated quantum dot display applications however the following points need to addressed before being accepted by the journal

- Fig. 2d presents an absorption enhancement factor (AEF) maxima of more than 30. The paper needs to scientifically elaborate the origin of the enhancement value here. What effects the maximal point here? What limits the enhancement factor?
- The fig 1 presented the stacked layer of alternating TiO₂ and SiO₂. (5 and 13 respectively) how did the authors decide on that? what determines the number of repeating units in here? Please explain. The same goes for the thickness of the alternating layers. How did the authors determine them as 49 nm and 77 nm; and what would be the effect of using thinner or thicker alternating layers?
- The extinction coefficient of the CQDs would definitely effect the result, especially the blue photon suppression. How would the results differ if perovskites or 2d nano platelets would be utilized rather than conventional QDs.
- - What is the not-normalized spectra look like? The color coordinate of the white light emission would be better if presented

Reviewer #3:

Remarks to the Author:

The paper introduces a novel approach for structurally engineered phosphors, utilizing a one-dimensional (1D) resonant cavity (RC) as a platform. Interestingly, the same author has previously explored engineered phosphors using a photonic crystal (PhC), as mentioned in references 17-19. What is notable is that while the author consistently compares the new results with a Reference Phosphor (Ref-PSP), there is no comparison to their previous work. This raises the question of why the author chose not to compare the new results with those obtained using

the photonic crystal.

In reference 17, the author asserts that "This observation indicates that the PhC phosphor is a viable technology for next-generation white LEDs and their applications." However, in the current paper, the author proposes a new platform for structurally engineered phosphors based on a resonant cavity. Once again, the comparison is made with the Reference Phosphor (Ref-PSP) rather than the previously investigated photonic crystal. Does this imply that the statement made in reference 17 is no longer valid and that the new resonant cavity platform (RC) is superior? It should be noted that the current work falls short of meeting the rigorous editorial criteria for broad impact and significance at Nature Communications. However, with the necessary revisions and corrections (pdf attached), it could still be of interest to a more specialized community.

Comment 1

In the paper, the author utilized a different approach by using a resonant cavity, to structurally engineer phosphors. The primary question is: What distinguishes this approach in terms of performance compared to the previously cited work (Ref19)?

Interestingly, both this paper and Ref19 compared same reference sample (Ref-PSP). It raises the question of why a performance comparison between this paper and Ref19 was not conducted. A novel study should ideally showcase advancements over prior research

Is the reference sample used (Ref-PSP) is a standard commercial sample, as mentioned in line 82-83 (ref 30)?

Comment 2

In lines 111-113 - compared to the conventional phosphor structure shown in Fig. 1a, the RC-PSP structure is only $\sim 1/10$ of its physical thickness and contains only $\sim 1/100$ of the net CQD amount.

“Where is the reference paper with the conventional phosphor structure that the author is talking about?”

Comment 3

Lines 118-119 - The extinction coefficient of the CQD film determined from independent spectroscopic ellipsometry measurements was used in the calculations.

Is it possible to include a reference or provide the measurements in the supplementary section?

Comment 4

Lines 119-123 - The peak absorbance and resonance linewidth for the RC-PSP were closely coupled and dependent on DRB layer numbers. Therefore, the appropriate RC structure that best fits given excitation requirements must be determined. This study chose the DBR layer numbers as $N_{ex} = 5$ (or 2.5 pairs) for 122 exDBR and $N_{em} = 13$ (or 6.5 pairs) for emDBR.

What criterion was used to select this specific number of pairs? Was an LED used as the excitation source? How would the device performance be affected if a laser diode was chosen instead, as mentioned here?

Comment 5

*Lines 177-180 - However, linewidth broadening is not necessarily a disadvantage because phosphors are typically excited by a light source with a broad bandwidth, such as an LED, for which an RC with a comparable resonance linewidth is preferable to **maximise** the use of excitation photons.*

However, in reference 34 that you cited, the use of a laser diode as the excitation source was mentioned. In this case, how would the broadening effect impact the performance of your device? Additionally, why is this point significant when you previously stated that phosphors are typically excited by a light source with a broad bandwidth? You also mentioned in lines 103-104 the use of the laser diode.

Comment 6

Lines 198-200 – The QD fluorescence intensities of RC-PSP and α RC-PSP significantly exceeded that of Ref-PSP.

Could you please provide the intensity values for the numbers? It is evident from the image that a comparison between RC-PSP and α RC-PSP with the reference Ref-PSP shows clear differences. However, when comparing RC-PSP with α RC-PSP, it is not as apparent.

Comment 7

Lines 216-218 - Hence, the high-index DBR material TiO₂ was replaced with Ta₂O₅ ($n_{\text{TaO}} \approx 2.08$), while the numbers of DBR layers were increased to $N_{\text{ex}} = 7$ (3.5 pairs) and $N_{\text{em}} = 31$ (15.5 pairs) to compensate for the lowered index contrast.

Will the number of DBR pairs significantly affect the device's cost in comparison to the initial RC-PSP design (6.5 to 15.5 pairs), considering its intended use in real-world applications?

Comment 8

Lines 229-231 - As shown in Figs. 3b and 3c, the FEFs are as high as ~6.8 and ~5.9 for the red and green RC-PSPs, respectively. Compared with the results shown in Fig. 2e, which were obtained with $\lambda_{ex} \approx 2 \text{ nm}$, the maximum FEFs are much lower, while the resonance linewidths become larger ($\lambda_0 \approx 20 \text{ nm}$).

The performance significantly decreased when an LED was used as the excitation source. How can this issue be addressed and resolved?

Comment 9

Line 288 – Fabrication of the RC-PSP

Please detail more the fabrication process: DBR thickness, growth rate, etc. The information should be available or a reference point it.

Last comment:

Given the concern over the toxicity of Cd, there has been a growing focus on developing environmentally friendly QLEDs that are free from heavy metals. Why should you continue to pursue this direction?

RESPONSE TO REVIEWER #1

► Comment 0

The work titled "Resonant cavity phosphor" by TAE-YUN LEE et al.. In this work one-dimensional resonant cavity that comprises a central cavity layer and two DBR mirrors on both sides as a platform for structurally engineered phosphors was fabricated and characterized. Three types of CQD phosphors were fabricated one without Bragg reflector and the other two phosphors fabricated with varying the thicknesses of Bragg layers. Overall, the work may be of interest to a broader audience. However, the authors should address the following points outlined below to improve the scientific quality. After carefully addressing the suggested revisions, this work may be considered for publication in the respected, Nature Communications Journal.

◄ Our Response

First of all, we the authors appreciate the time and effort that the reviewer has spent for reviewing our manuscript. We also thank the reviewer for expressing favorable opinions on our work. We have done our best in responding to the reviewer's comments. We hope that our responses resolve the reviewer's concerns satisfactorily.

► Comment 1

The authors should add details about the bare CQD synthesis procedure or add references that include the synthesis information.

◄ Our Response

The CQDs used in our experiments were procured commercially, as clarified in the Methods section in our manuscript. The detailed CQD synthesis procedure is therefore the company's proprietary information so that we cannot inquire of the company about it. Nonetheless, an engineer of the company mentioned to us that their CQDs are of standard CdSe-ZnS core-shell structure, which is also clarified in the Methods section. The references for the synthesis and characterizations of the CdSe-ZnS CQDs are abundant in the literature; for example, Reference

6 in our manuscript should be a good one. To make this circumstance clearer, we have revised our manuscript as follows:

[Before]

“Subsequently, CdSe-ZnS core-shell CQDs (CZO-620H and CZO-530H, Zeus) dispersed in a cyclohexane solution were spin-coated on top of the first SiO₂ wing layer.”

[After]

“Subsequently, commercially procured CdSe-ZnS core-shell CQDs (CZO-620H and CZO-530H, Zeus) were dispersed in a cyclohexane solution and spin-coated on top of the first SiO₂ wing layer.”

► Comment 2

The fluorescence spectra of the quantum dots should be added to the text or Supplementary Information.

◀ Our Response

We would like to remind the reviewer that the intrinsic fluorescence (FL) spectra for the red and green CQDs used in our experiments are shown in Figs. 3b and 3c, respectively; they are even shaded in the corresponding red and green colours for clarity.

Although somewhat indirect, the intrinsic FL spectrum for the red CQDs is also seen in the PLE data for Ref-PSP—Fig. 2b-(i).

► Comment 3

The symbol DRB line 120, page 6, should be corrected.

◄ Our Response

The authors thank the reviewer for finding the typo. We have corrected it in our revised manuscript.

► Comment 4

The results of absorption calculation showed that the absorption maximum located at 450 nm, and the excitation maximum located at 468 nm, whereas the emission located in the red region with high Stokes shift. These results indicate that the emission from quantum dots originating from trapping state and this contradict with the core-shell structure, could the authors explain?

◄ Our Response

We the authors are afraid that we do not have clear understandings on the reviewer's points. Specifically, the definition of 'trapping state' and the exact meaning of the subsequent question are still puzzling to us. In our reply below, we assume that the reviewer meant 'quantum

confined states' inside the core-shell CQD structure with the 'trapping state'. If we misunderstood, please let us know and we will happily answer again.

First of all, we would like to remind the reviewer that the operation condition of our RC-PSP is that the phosphor excitation wavelength is tuned to the resonant wavelength of RC: $\lambda_{\text{ex}} = \lambda_0$. Therefore, there are only two wavelengths that are distinguished: one is the resonant wavelength (λ_0) of the RC-PSP, which is equivalent to the excitation wavelength (λ_{ex}), and the other is the intrinsic CQD emission wavelength (λ_{R} or λ_{G}). Although our RC-PSP was designed such that its resonance should occur at 450 nm—Fig. 2a-(ii) & (iii), it turned out to be 468 nm in the fabricated devices as confirmed by the PLE experiments—Fig. 2b-(ii) & (iii). The resultant 18 nm shift is a discrepancy between the ideality and reality, which can be hardly avoided. Consequently, we were obliged to excite our RC-PSP at the experimentally confirmed resonant wavelength (468 nm) and obtained fluorescence at the intrinsic CQD emission wavelength (620/530 nm).

Now, we are in the position to answer to the question raised in the second sentence. But we do NOT see any contradiction. In case that the reviewer concerns about the large Stokes shift between the absorption and emission, it is simply due to the fact that semiconductor band structures are continuous, which differs from traditional phosphors that rely on metal ion transitions between discrete energy levels. Consequently, resultant photonic density of states is continuous in its distribution, inferring that CQDs can be excited at any wavelength shorter than CQD emission wavelength. Therefore, the excitation/absorption wavelength is somewhat arbitrary, and a large Stokes shift may happen. In the present study, for example, we chose the blue (450 nm in design; 468 nm in experiments) for the excitation of green (530 nm) and red (620 nm) CQDs with the R/G/B full-colour display in mind, resulting in a huge separation between excitation and emission. If we are in a completely wrong place far away from the reviewer's point, then please guide us.

RESPONSE TO REVIEWER #2

► Comment 0

In this work Jeon et. al. proposes a one dimensional resonant cavity, including a cavity layer and two DBR mirrors, for structurally engineered phosphor demonstration.

The work presented is a simple yet an interesting approach to tailor the overall light output for pixelated color converters with a Q factor of around 90. The results are believed to open up new applications and designs for the pixelated quantum dot display applications however the following points need to addressed before being accepted by the journal

◄ Our Response

We the authors thank the reviewer for the positive opinions on our work. We hope that our responses below could resolve the reviewer's remaining concerns.

► Comment 1

Fig. 2d presents an absorption enhancement factor (AEF) maxima of more than 30. The paper needs to scientifically elaborate the origin of the enhancement value here. What effects the maximal point here? What limits the enhancement factor?

◄ Our Response

We are very sorry if the reviewer feels that we were not elaborate enough in explaining the origin of the absorption enhancement. As we already mentioned in the introductory part of the manuscript, however, the absorption by an electric dipole (CQD in the present case) can be enhanced by putting it inside RC where photonic density of states (or electric field strength) is significantly amplified due to RC effect. Enhancement in absorption, on the other hand, depends on the detailed design of RC, especially the reflectivity of each DBR (or the number of dielectric layers in each DBR mirror). Generally speaking, the higher the DBR reflectivity (or the more layers in DBRs), the sharper the resonance and the higher the maximum absorbance. These are only qualitative explanations, and the exact absorption spectrum can be obtained by numerical calculations, which is exactly what we have shown in Supplementary

Information S1 for various combinations of the DBR layer numbers (N_{em} , N_{ex}). The enhancement factor can then be obtained by taking the absorbance ratio between RC-PSP and Ref-PSP, as explained in the manuscript:

“The absorption enhancement factor (AEF), defined as the absorbance ratio between RC-PSP and Ref-PSP, was deduced as a function of wavelength, as shown in Fig. 2d.”

► Comment 2

The fig 1 presented the stacked layer of alternating TiO₂ and SiO₂. (5 and 13 respectively) how did the authors decide on that? what determines the number of repeating units in here? Please explain. The same goes for the thickness of the alternating layers. How did the authors determine them as 49 nm and 77 nm; and what would be the effect of using thinner or thicker alternating layers?

◄ Our Response

First of all, the thickness of each layer in the DBRs is determined by the standard design rule for DBR: $d = \lambda/4n$. This condition is clearly described in the manuscript as follows:

“The two DBRs consist of alternating $\lambda/4$ -thick dielectric layers of TiO₂ ($d_{TiO_2} \approx 49$ nm, $n_{TiO_2} \approx 2.32$) and SiO₂ ($d_{SiO_2} \approx 77$ nm, $n_{SiO_2} \approx 1.47$), starting and ending with the TiO₂ layers for the higher index contrasts to the environmental materials, air and silica.”

Based on the description, it is easy and straightforward to deduce the thickness values for the TiO₂ and SiO₂ layers: $d_{TiO_2} = \lambda_0 / (4 \cdot n_{TiO_2}) = (450 \text{ nm}) / (4 \times 2.32) \approx 49$ nm and $d_{SiO_2} = \lambda_0 / (4 \cdot n_{SiO_2}) = (450 \text{ nm}) / (4 \times 1.47) \approx 77$ nm.

If the thicknesses of the two kinds of DBR layers are varied in proportion, the centre wavelength of the resultant DBR stopband will be rescaled accordingly. If they are varied independently, then results can be quite complex, but can still be calculated. Because our RC design stuck to the standard DBRs, however, we did not consider thicknesses other than $d = \lambda/4n$.

Regarding the reviewer’s question on how we determined the layer numbers, it is explained in detail in Supplementary Information S2. In short, the chosen combination ($N_{emDBR} + N_{exDBR} = 5 + 13$) offers high values for all the three figure-of-merits in absorption with a relatively small number of the total DBR layers: peak absorbance (suitable for a laser-like sharp

excitation source), integrated absorbance (suitable for an excitation source with a flat intensity profile), and weighted absorbance (suitable for a LED-like broad excitation source). Therefore, our choice is a compromised solution to make our RC-PSP adaptable broadly to various kinds of excitation sources from LD-like to LED-like ones.

To make clear that S2, which has become S3 in the revised manuscript, deals with the present issue, we have revised its title as follows:

[Before]

“S3. Absorbance characteristics of the RC-PSP”

[After]

“S3. How to determine the DBR layer numbers in the RC-PSP”

We have also added a sentence in the figure caption of S3 to clarify the practical implications of the three different absorbances as follows:

[Added]

“Note that the peak, integrated, and weighted absorbances represent the figure-of-merits appropriate for laser-like, broad-emission-bandwidth, and LED-like excitation sources, respectively.”

► Comment 3

The extinction coefficient of the CQDs would definitely effect the result, especially the blue photon suppression. How would the results differ if perovskites or 2d nano platelets would be utilized rather than conventional QDs.

◀ Our Response

Thanks for the interesting suggestion! As we commented in the manuscript, the RC-PSP is a structurally engineered phosphor, thus ‘not bound to any specific phosphor material.’ In this context, perovskites or 2D nanoplatelets—or anything else—can be also employed as long as they can be physically incorporated into the RC structure, i.e., as long as they can be cast into a thin film of an appropriate thickness ($d = \lambda/2n$) with flat smooth surfaces. If the resultant film has a higher/lower extinction coefficient than CQDs, then a less/greater number of DBR layers would be required in order to obtain similar performance. The optimum RC structure for the corresponding phosphor material can be obtained through thorough absorbance calculations as done for CQDs in the present study.

▶ Comment 4

What is the not-normalized spectra look like? The color coordinate of the white light emission would be better if presented.

◀ Our Response

As pointed out in the manuscript, we have identified full-colour display as the main application sector of the RC-PSP, although white light generation is also possible. For the generation of white colour with good colour coordinates—presumably close to (0.333, 0.333), the three primary colours (R/G/B) should be mixed delicately and correctly in terms of both wavelength and intensity, which would be a formidable task that requires substantial time and efforts.

On the contrary, the full-colour display only requires independent three primary colour sources with high colour purity. All that matters here is the centre wavelengths (for R/G/B colours) and associated spectral linewidths; relative R/G/B intensities required to render a certain colour can be attained by controlling individual colour sources independently. In other words, there is no need to care about the relative intensities of R/G/B! In fact, when the primary colour coordinates (and the corresponding chromaticity colour triangle) of a given display

system are evaluated, three emission spectra from the R/G/B sources are intensity-normalized beforehand, as done in the present study.

To clarify that efficient white light generation could be another important application area for our RC-PSP as the reviewer pointed out, we have revised our manuscript as shown below:

[Before]

“The most obvious and impactful application sector for RC-PSP is the development of full-colour displays based on hybrid micro-LED arrays, in which independently addressable red-green-blue (RGB) pixels can be prepared by combining blue LEDs with red and green phosphors³⁰, as depicted in Fig. 3a.”

[After]

“Although efficient white light generation is certainly a possibility, the most obvious and impactful application sector for RC-PSP should be the development of full-colour displays based on hybrid micro-LED arrays, in which independently addressable red-green-blue (RGB) pixels can be prepared by combining blue LEDs with red and green phosphors³⁰, as depicted in Fig. 3a.”

RESPONSE TO REVIEWER #3

► Comment 0

The paper introduces a novel approach for structurally engineered phosphors, utilizing a one-dimensional (1D) resonant cavity (RC) as a platform. Interestingly, the same author has previously explored engineered phosphors using a photonic crystal (PhC), as mentioned in references 17-19. What is notable is that while the author consistently compares the new results with a Reference Phosphor (Ref-PSP), there is no comparison to their previous work. This raises the question of why the author chose not to compare the new results with those obtained using the photonic crystal.

In reference 17(Adv. Mater. 30, 1703506 (2018)), the author asserts that "This observation indicates that the PhC phosphor is a viable technology for next-generation white LEDs and their applications." However, in the current paper, the author proposes a new platform for structurally engineered phosphors based on a resonant cavity. Once again, the comparison is made with the Reference Phosphor (Ref-PSP) rather than the previously investigated photonic crystal. Does this imply that the statement made in reference 17 is no longer valid and that the new resonant cavity platform (RC) is superior?

It should be noted that the current work falls short of meeting the rigorous editorial criteria for broad impact and significance at Nature Communications. However, with the necessary revisions and corrections (pdf attached), it could still be of interest to a more specialized community.

◄ Our Response

As the reviewer pointed out, we did not compare the RC phosphor with the PhC phosphor, which is the original version of the structurally engineered phosphor we developed. It is simply because the RC phosphor is far more superior than the PhC phosphor in both structure and performance. Details are thoroughly explained below. We hope that our replies are satisfactory in the reviewer's high standards.

► Comment 1

In the paper, the author utilized a different approach by using a resonant cavity, to structurally engineer phosphors. The primary question is: What distinguishes this approach in terms of performance compared to the previously cited work (Ref19)?

Interestingly, both this paper and Ref19 compared same reference sample (Ref-PSP). It raises the question of why a performance comparison between this paper and Ref19 was not conducted. A novel study should ideally showcase advancements over prior research

Is the reference sample used (Ref-PSP) is a standard commercial sample, as mentioned in line 82-83 (ref 30)?

◀ Our Response

Let me outline the brief history of how the structurally engineered phosphor project has evolved in our lab. As a group working on photonic crystals for a long time, we got an idea of how to improve phosphor performance via PhC band-edge modes, which marks the beginning of the structurally engineered phosphor research. Since then, we have been upgrading the PhC phosphor continuously and also seeking for its possible applications, as witnessed by a series of publications (Ref. 16–19). Quite lately, we realized that RC can also play a similar role for phosphor. Further, RC is more attractive than PhC because RC can be fabricated layer-by-layer, which greatly facilitates the realization of the platform and thus strongly appeal for real applications. In good contrast, PhC requires a sophisticated lateral submicron-patterning process, making its practical applications quite challenging. To make a long story short, the RC-PSP is a totally different platform from the PhC phosphor so that we did not and still do not think the direct comparisons between them are really necessary in the present manuscript that deals with the RC-PSP.

As for their performances, it turns out that the RC-PSP performs far better than the PhC phosphor. For example, the RC-PSP used specifically in the present study offers a high absorbance $A \approx 0.85$ —which can be improved even further by increasing cavity Q, whereas the PhC phosphor described in our latest research only exhibits $A \approx 0.5$. All the other performance results, both simulations and experiments, also confirmed the superiority of the RC-PSP. All these can be easily confirmed by comparing the results described in the present manuscript with those of Reference 19.

To clarify the superiority of RC-PSP over PhC phosphor, we have added a paragraph in the revised manuscript as follows:

[Added]

“Furthermore, the RC-PSP is superior to the PhC phosphor in terms of both structure and performance. The RC-PSP can be constructed by vertically stacking planar layers one after another, which is far easier than the sophisticated lateral submicron-patterning process required for fabricating the PhC phosphor. As for the performance, the absorption of excitation photons is greater in the RC-PSP than in the PhC phosphor as revealed hereafter.”

Regarding the reference, on the other hand, we believe that the fair form of the reference phosphor for any structurally engineered phosphor should be the one without any structure (or a simple thin film), but prepared in a similar manner (spin-coating) and containing the same amount of CQDs. This principle was also applied to our previous researches on the PhC phosphors. The reference phosphor for the present study is the cavity layer itself, as described in the manuscript, which differs from that of the previous PhC phosphor—in terms of the CQD amount (or the CQD film thickness). Obviously, it also differs from the one shown in Fig. 1a (i.e., CQDs dispersed in a thick polymer film, which is used in the conventional display colour pixels)—in terms of the structural configuration.

To elucidate the Ref-PSP structure used in the present study more clearly, the manuscript has been revised as follows:

[Before]

“Conversely, the reference phosphor (Ref-PSP), which corresponds to the cavity layer itself—including the two SiO₂ wing layers, exhibits a fairly flat spectral profile with no resonance feature, and the absorbance at $\lambda_0 = 450$ nm is only $A \approx 0.03$ —Fig. 2a(i).”

[After]

“Conversely, the reference phosphor (Ref-PSP), which corresponds to the cavity layer itself—a 40-nm-thick CQD layer cladded by the two SiO₂ wing layers on both sides, exhibits a fairly flat spectral profile with no resonance feature, and the absorbance at $\lambda_0 = 450$ nm is only $A \approx 0.03$ —Fig. 2a(i).”

► Comment 2

In lines 111-113 - compared to the conventional phosphor structure shown in Fig. 1a, the RC-PSP structure is only $\sim 1/10$ of its physical thickness and contains only $\sim 1/100$ of the net CQD amount.

“Where is the reference paper with the conventional phosphor structure that the author is talking about?”

◀ Our Response

Please accept our apology for omitting a proper reference. Our claims in the sentence are based on Ref. 32 (also confirmed by private conversations with the authors of the reference). The manuscript has been revised as follows:

[Before]

“It should be noted that, compared to the conventional phosphor structure shown in Fig. 1a, the RC-PSP structure is only $\sim 1/10$ of its physical thickness and contains only $\sim 1/100$ of the net CQD amount.”

[After]

“It should be noted that, compared to the conventional phosphor structure shown in Fig. 1a, the RC-PSP structure is only $\sim 1/10$ of its physical thickness and contains only $\sim 1/100$ of the net CQD amount³².”

▶ Comment 3

Lines 118-119 - The extinction coefficient of the CQD film determined from independent spectroscopic ellipsometry measurements was used in the calculations.

Is it possible to include a reference or provide the measurements in the supplementary section?

◀ Our Response

We agree that the information on the extinction coefficients (or the entire dispersions of the complex refractive index) of the CQD films should be helpful for readers to understand our work better. To comply with the reviewer’s suggestion, we have added a section in Supplementary Information to provide the n & k dispersions of the both red and green CQD films used in our experiments.

[Added]

S2. Complex refractive indices of the CQD films

Fig. S2 | Dispersion relations of n and k of the CQD films used in the experiments. n and k measured as functions of wavelength by spectroscopic ellipsometry technique are presented for **a**, the red CQD film and **b**, the green CQD film.

► Comment 4

Lines 119-123 - The peak absorbance and resonance linewidth for the RC-PSP were closely coupled and dependent on DRB layer numbers. Therefore, the appropriate RC structure that best fits given excitation requirements must be determined. This study chose the DBR layer numbers as $N_{\text{ex}} = 5$ (or 2.5 pairs) for 122 exDBR and $N_{\text{em}} = 13$ (or 6.5 pairs) for emDBR.

What criterion was used to select this specific number of pairs? Was an LED used as the excitation source? How would the device performance be affected if a laser diode was chosen instead, as mentioned here?

◀ Our Response

The question on the DBR layer numbers was also asked by Reviewer #2 in his/her Comment 2. Once again, Supplementary Information S2 explains how the DBR layer numbers used in the present study are determined. To repeat our answer again, the combination we chose ($N_{\text{emDBR}} + N_{\text{exDBR}} = 5 + 13$) offers high values for all the three figure-of-merits with a relatively small number of the total DBR layers: peak absorbance (suitable for a laser-like excitation source), integrated absorbance (suitable for an excitation source with a constant intensity

profile), and weighted absorbance (suitable for a LED-like excitation source). Therefore, our choice is a compromised solution to make our RC-PSP adaptable broadly to various kinds of excitation sources from LD-like to LED-like ones.

To make clear that S2, which has become S3 in the revised manuscript, deals with the present issue, we have revised its title as follows:

[Before]

“S3. Absorbance characteristics of the RC-PSP”

[After]

“S3. How to determine the DBR layer numbers in the RC-PSP”

We have also added a sentence to the figure caption of S3 to clarify the practical implications of the three different absorbances as follows:

[Added]

“Note that the peak, integrated, and weighted absorbances may represent the figure-of-merits appropriate for LD-like, broad-emission-bandwidth, and LED-like excitation sources, respectively.”

Regarding the excitation source used in our experiments, which is schematically illustrated in the inset of Fig. 2b (i), it is neither LED nor LD, but somewhere between the two. It is LED-like because it has a broad emission bandwidth (~20 nm), and at the same time it is also LD-like because it is collimated.

Lastly, if LD is used for excitation, the RC resonance can be intentionally designed to be even sharper to obtain higher absorption of excitation photons. However, this condition makes it a formidable task to precisely tune the LD wavelength to the RC resonance peak. We have already made a short comment on this in Ref. 34.

► Comment 5

Lines 177-180 (aRC res. broadening) - However, linewidth broadening is not necessarily a disadvantage because phosphors are typically excited by a light source with a broad bandwidth, such as an LED, for which an RC with a comparable resonance linewidth is preferable to maximise the use of excitation photons.

However, in reference 34 that you cited, the use of a laser diode as the excitation source was mentioned. In this case, how would the broadening effect impact the performance of your device? Additionally, why is this point significant when you previously stated that phosphors are typically excited by a light source with a broad bandwidth? You also mentioned in lines 103-104 the use of the laser diode.

◄ Our Response

In principle, a LD-like excitation source, characterized by a sharp emission linewidth (typically less than 1 nm), should be ideal for our RC-PSP. RC can then be designed such that its resonance matches exactly with the LD emission in terms of both wavelength and linewidth to induce near 100% absorption of excitation photons. In practice, however, the exact wavelength tuning becomes extremely difficult when the both linewidths (LD emission and RC resonance) are narrow. Meanwhile, most of the phosphor excitation sources employed in real and practical devices are LEDs as mentioned in our manuscript, presumably because of economic reasons. In the view point of operating the RC-PSP, a LED-like excitation source makes the wavelength tuning far easier because its linewidth is much broader (> 10 nm). It is in this context that we said in our manuscript that a broadened resonance linewidth is not

necessarily a disadvantage. Therefore, characteristics of a given excitation source should/could be properly reflected in the RC design, for example, by choosing a suitable figure-of-merit among the maximum/integrated/weighted absorbances.

► Comment 6

Lines 198-200 (camera image) – The CQD fluorescence intensities of RC-PSP and α RC-PSP significantly exceeded that of Ref-PSP.

Could you please provide the intensity values for the numbers? It is evident from the image that a comparison between RC-PSP and α RC-PSP with the reference Ref-PSP shows clear differences. However, when comparing RC-PSP with α RC-PSP, it is not as apparent.

◀ Our Response

The intensity information that the reviewer inquired can be found in Fig. 2b, where detailed PLE data are shown in 3D format.

Considering our manuscript in the reviewer's (or readers') point of view, however, the authors have realized that the quantitative fluorescence intensity comparisons among Ref-PSP, RC-PSP, and α RC-PSP are difficult to make from both the PLE data (Fig. 2b) and the

photograph images (Fig. 2c). In order to facilitate such quantitative comparisons and also to comply with the reviewer’s request, we have added a table along with a sentence to explain about it in the revised manuscript.

[Added]

“The QD fluorescence intensities of RC-PSP and α RC-PSP significantly exceeded that of Ref-PSP. For quantitative comparisons, both the peak and integrated fluorescence intensities for Ref-PSP, RC-PSP, and α RC-PSP are extracted from the PLE data in Fig. 2b and summarized in Table 1.”

Table 1. Peak and integrated fluorescence intensities for Ref-PSP, RC-PSP, and α RC-PSP

	Ref-PSP	RC-PSP	α RC-PSP
Peak Intensity	190	6,070	7,220
Integrated Intensity	6,530	191,650	209,720

*The numbers are deduced from the PLE data in Fig. 2b.

► Comment 7

Lines 216-218 - Hence, the high-index DBR material TiO₂ was replaced with Ta₂O₅ (n_{TaO} ≈ 2.08), while the numbers of DBR layers were increased to N_{ex} = 7 (3.5 pairs) and N_{em} = 31 (15.5 pairs) to compensate for the lowered index contrast.

Will the number of DBR pairs significantly affect the device's cost in comparison to the initial RC-PSP design (6.5 to 15.5 pairs), considering its intended use in real-world applications?

◀ Our Response

As clarified in the Methods section in the manuscript, the DBRs are vacuum-deposited using e-gun evaporator. Due to the nature of vacuum process, it is charged by runs, not by the number of layers. In other words, an increase in the DBR layer numbers would not affect the cost at all.

► Comment 8

Lines 229-231 - As shown in Figs. 3b and 3c, the FEFs are as high as ~ 6.8 and ~ 5.9 for the red and green RC-PSPs, respectively. Compared with the results shown in Fig. 2e, which were obtained with $\lambda_{\text{ex}} \approx 2 \text{ nm}$, the maximum FEFs are much lower, while the resonance linewidths become larger ($\lambda_0 \approx 20 \text{ nm}$).

The performance significantly decreased when an LED was used as the excitation source. How can this issue be addressed and resolved?

◀ Our Response

This issue was already discussed in our response to Comment 5. In short, the narrower the emission linewidth of excitation source and the resonant linewidth of RC, the better the RC-PSP performance. This argument is consistent with our experimental results performed with two different excitation linewidths: 2 nm and 20 nm. This is an intrinsic effect, implying that there is nothing we can do about it, but we have to live with it. Nonetheless, please note that the fluorescence enhancement factors of ~ 6.8 and ~ 5.9 are already quite impressive numbers! In addition, we would like to remind the reviewer that our RC-PSP structure is a compromised design for both LD-like and LED-like excitation sources, as explained in our Response to Comment 4. Therefore, we could redesign our RC for optimised or best performance when the excitation source is fixed and its characteristics are given.

▶ Comment 9

Line 288 – Fabrication of the RC-PSP

Please detail more the fabrication process: DBR thickness, growth rate, etc. The information should be available or a reference point it.

◀ Our Response

We believe that the RC-PSP fabrication details are already well described in the Methods section, which is copied below.

Fabrication of the RC-PSP

Device fabrication involved three simple steps: two vacuum depositions and one spin-coating in between. First, the $\text{TiO}_2/\text{SiO}_2$ (or $\text{Ta}_2\text{O}_5/\text{SiO}_2$) emDBR and the first SiO_2 wing layer were deposited on a fused quartz substrate using an e-gun evaporator at an elevated temperature of

T = 150°C. During the deposition of the high-index layers, an O₂ environment was provided to fully oxidise the metal ions. Subsequently, CdSe-ZnS core-shell CQDs (CZO-620H and CZO-530H, Zeus) dispersed in a cyclohexane solution were spin-coated on top of the first SiO₂ wing layer. The CQD concentration (2 wt. %) and spin speed (3000 rpm) were carefully selected to obtain the desired CQD film thickness (40 nm). Finally, another round of vacuum deposition of the second SiO₂ wing layer and exDBR was performed to complete the device fabrication. The first and last layers of both emDBR and exDBR was a high-index layer (TiO₂ or Ta₂O₅). For α RC-PSP, exDBR and α DBR were simultaneously deposited in a single run.

The information on the DBR layer thicknesses can be found in the main body of the manuscript, which is also copied below.

The two DBRs consist of alternating $\lambda/4$ -thick dielectric layers of TiO₂ ($d_{\text{TiO}_2} \approx 49$ nm, $n_{\text{TiO}_2} \approx 2.32$) and SiO₂ ($d_{\text{SiO}_2} \approx 77$ nm, $n_{\text{SiO}_2} \approx 1.47$), starting and ending with the TiO₂ layers for the higher index contrasts to the environmental materials, air and silica.

The only information that the reviewer specified but we cannot provide is the growth rate (or deposition rate) of the DBR layers because it is the proprietary information of the company from which we outsourced the vacuum depositions. Nonetheless, a typical deposition rate for dielectric films in e-gun evaporator is a few Å/s. We are sure that the deposition rates the company used for our samples is also within the range. If the reviewer still feels that further details should be provided, please be specific and we will happily answer.

► Last Comment

Given the concern over the toxicity of Cd, there has been a growing focus on developing environmentally friendly QLEDs that are free from heavy metals. Why should you continue to pursue this direction?

◀ Our Response

We are well aware that the use of Cd-based CQDs are now tightly controlled due to ever-growing environmental issues and even strictly forbidden in commercial products. However, the Cd-free CQDs, such as InP-based ones, are not only expensive but also difficult to procure,

whereas Cd-based conventional CQDs are readily available commercially at an affordable price. Naturally, university labs have been still using Cd-based CQDs for R&D purposes and in small volumes. Please be reminded that the concept of the structurally engineered phosphors is not bound to Cd-based CQDs but applicable to any colour-converting materials. It is only because of the availability and affordability that we have stuck to Cd-based CQDs.

===== End of Report =====

Reviewers' Comments:

Reviewer #1:

Remarks to the Author:

Accept

Reviewer #2:

Remarks to the Author:

The authors have addressed all queries scientifically during the revision, therefore from my perspective the paper can now be accepted for publication.

Reviewer #3:

None

RESPONSE TO REVIEWER #1

► Comment 0

The work titled "Resonant cavity phosphor" by TAE-YUN LEE et al.. In this work one-dimensional resonant cavity that comprises a central cavity layer and two DBR mirrors on both sides as a platform for structurally engineered phosphors was fabricated and characterized. Three types of CQD phosphors were fabricated one without Bragg reflector and the other two phosphors fabricated with varying the thicknesses of Bragg layers. Overall, the work may be of interest to a broader audience. However, the authors should address the following points outlined below to improve the scientific quality. After carefully addressing the suggested revisions, this work may be considered for publication in the respected, Nature Communications Journal.

◄ Our Response

First of all, we the authors appreciate the time and effort that the reviewer has spent for reviewing our manuscript. We also thank the reviewer for expressing favorable opinions on our work. We have done our best in responding to the reviewer's comments. We hope that our responses resolve the reviewer's concerns satisfactorily.

► Comment 1

The authors should add details about the bare CQD synthesis procedure or add references that include the synthesis information.

◄ Our Response

The CQDs used in our experiments were procured commercially, as clarified in the Methods section in our manuscript. The detailed CQD synthesis procedure is therefore the company's proprietary information so that we cannot inquire of the company about it. Nonetheless, an engineer of the company mentioned to us that their CQDs are of standard CdSe-ZnS core-shell structure, which is also clarified in the Methods section. The references for the synthesis and characterizations of the CdSe-ZnS CQDs are abundant in the literature; for example, Reference

6 in our manuscript should be a good one. To make this circumstance clearer, we have revised our manuscript as follows:

[Before]

“Subsequently, CdSe-ZnS core-shell CQDs (CZO-620H and CZO-530H, Zeus) dispersed in a cyclohexane solution were spin-coated on top of the first SiO₂ wing layer.”

[After]

“Subsequently, commercially procured CdSe-ZnS core-shell CQDs (CZO-620H and CZO-530H, Zeus) were dispersed in a cyclohexane solution and spin-coated on top of the first SiO₂ wing layer.”

► Comment 2

The fluorescence spectra of the quantum dots should be added to the text or Supplementary Information.

◀ Our Response

We would like to remind the reviewer that the intrinsic fluorescence (FL) spectra for the red and green CQDs used in our experiments are shown in Figs. 3b and 3c, respectively; they are even shaded in the corresponding red and green colours for clarity.

Although somewhat indirect, the intrinsic FL spectrum for the red CQDs is also seen in the PLE data for Ref-PSP—Fig. 2b-(i).

► Comment 3

The symbol DRB line 120, page 6, should be corrected.

◄ Our Response

The authors thank the reviewer for finding the typo. We have corrected it in our revised manuscript.

► Comment 4

The results of absorption calculation showed that the absorption maximum located at 450 nm, and the excitation maximum located at 468 nm, whereas the emission located in the red region with high Stokes shift. These results indicate that the emission from quantum dots originating from trapping state and this contradict with the core-shell structure, could the authors explain?

◄ Our Response

We the authors are afraid that we do not have clear understandings on the reviewer's points. Specifically, the definition of 'trapping state' and the exact meaning of the subsequent question are still puzzling to us. In our reply below, we assume that the reviewer meant 'quantum

confined states' inside the core-shell CQD structure with the 'trapping state'. If we misunderstood, please let us know and we will happily answer again.

First of all, we would like to remind the reviewer that the operation condition of our RC-PSP is that the phosphor excitation wavelength is tuned to the resonant wavelength of RC: $\lambda_{\text{ex}} = \lambda_0$. Therefore, there are only two wavelengths that are distinguished: one is the resonant wavelength (λ_0) of the RC-PSP, which is equivalent to the excitation wavelength (λ_{ex}), and the other is the intrinsic CQD emission wavelength (λ_{R} or λ_{G}). Although our RC-PSP was designed such that its resonance should occur at 450 nm—Fig. 2a-(ii) & (iii), it turned out to be 468 nm in the fabricated devices as confirmed by the PLE experiments—Fig. 2b-(ii) & (iii). The resultant 18 nm shift is a discrepancy between the ideality and reality, which can be hardly avoided. Consequently, we were obliged to excite our RC-PSP at the experimentally confirmed resonant wavelength (468 nm) and obtained fluorescence at the intrinsic CQD emission wavelength (620/530 nm).

Now, we are in the position to answer to the question raised in the second sentence. But we do NOT see any contradiction. In case that the reviewer concerns about the large Stokes shift between the absorption and emission, it is simply due to the fact that semiconductor band structures are continuous, which differs from traditional phosphors that rely on metal ion transitions between discrete energy levels. Consequently, resultant photonic density of states is continuous in its distribution, inferring that CQDs can be excited at any wavelength shorter than CQD emission wavelength. Therefore, the excitation/absorption wavelength is somewhat arbitrary, and a large Stokes shift may happen. In the present study, for example, we chose the blue (450 nm in design; 468 nm in experiments) for the excitation of green (530 nm) and red (620 nm) CQDs with the R/G/B full-colour display in mind, resulting in a huge separation between excitation and emission. If we are in a completely wrong place far away from the reviewer's point, then please guide us.

RESPONSE TO REVIEWER #2

► Comment 0

In this work Jeon et. al. proposes a one dimensional resonant cavity, including a cavity layer and two DBR mirrors, for structurally engineered phosphor demonstration.

The work presented is a simple yet an interesting approach to tailor the overall light output for pixelated color converters with a Q factor of around 90. The results are believed to open up new applications and designs for the pixelated quantum dot display applications however the following points need to addressed before being accepted by the journal

◄ Our Response

We the authors thank the reviewer for the positive opinions on our work. We hope that our responses below could resolve the reviewer's remaining concerns.

► Comment 1

Fig. 2d presents an absorption enhancement factor (AEF) maxima of more than 30. The paper needs to scientifically elaborate the origin of the enhancement value here. What effects the maximal point here? What limits the enhancement factor?

◄ Our Response

We are very sorry if the reviewer feels that we were not elaborate enough in explaining the origin of the absorption enhancement. As we already mentioned in the introductory part of the manuscript, however, the absorption by an electric dipole (CQD in the present case) can be enhanced by putting it inside RC where photonic density of states (or electric field strength) is significantly amplified due to RC effect. Enhancement in absorption, on the other hand, depends on the detailed design of RC, especially the reflectivity of each DBR (or the number of dielectric layers in each DBR mirror). Generally speaking, the higher the DBR reflectivity (or the more layers in DBRs), the sharper the resonance and the higher the maximum absorbance. These are only qualitative explanations, and the exact absorption spectrum can be obtained by numerical calculations, which is exactly what we have shown in Supplementary

Information S1 for various combinations of the DBR layer numbers (N_{em} , N_{ex}). The enhancement factor can then be obtained by taking the absorbance ratio between RC-PSP and Ref-PSP, as explained in the manuscript:

“The absorption enhancement factor (AEF), defined as the absorbance ratio between RC-PSP and Ref-PSP, was deduced as a function of wavelength, as shown in Fig. 2d.”

► Comment 2

The fig 1 presented the stacked layer of alternating TiO₂ and SiO₂. (5 and 13 respectively) how did the authors decide on that? what determines the number of repeating units in here? Please explain. The same goes for the thickness of the alternating layers. How did the authors determine them as 49 nm and 77 nm; and what would be the effect of using thinner or thicker alternating layers?

◄ Our Response

First of all, the thickness of each layer in the DBRs is determined by the standard design rule for DBR: $d = \lambda/4n$. This condition is clearly described in the manuscript as follows:

“The two DBRs consist of alternating $\lambda/4$ -thick dielectric layers of TiO₂ ($d_{TiO_2} \approx 49$ nm, $n_{TiO_2} \approx 2.32$) and SiO₂ ($d_{SiO_2} \approx 77$ nm, $n_{SiO_2} \approx 1.47$), starting and ending with the TiO₂ layers for the higher index contrasts to the environmental materials, air and silica.”

Based on the description, it is easy and straightforward to deduce the thickness values for the TiO₂ and SiO₂ layers: $d_{TiO_2} = \lambda_0 / (4 \cdot n_{TiO_2}) = (450 \text{ nm}) / (4 \times 2.32) \approx 49$ nm and $d_{SiO_2} = \lambda_0 / (4 \cdot n_{SiO_2}) = (450 \text{ nm}) / (4 \times 1.47) \approx 77$ nm.

If the thicknesses of the two kinds of DBR layers are varied in proportion, the centre wavelength of the resultant DBR stopband will be rescaled accordingly. If they are varied independently, then results can be quite complex, but can still be calculated. Because our RC design stuck to the standard DBRs, however, we did not consider thicknesses other than $d = \lambda/4n$.

Regarding the reviewer’s question on how we determined the layer numbers, it is explained in detail in Supplementary Information S2. In short, the chosen combination ($N_{emDBR} + N_{exDBR} = 5 + 13$) offers high values for all the three figure-of-merits in absorption with a relatively small number of the total DBR layers: peak absorbance (suitable for a laser-like sharp

excitation source), integrated absorbance (suitable for an excitation source with a flat intensity profile), and weighted absorbance (suitable for a LED-like broad excitation source). Therefore, our choice is a compromised solution to make our RC-PSP adaptable broadly to various kinds of excitation sources from LD-like to LED-like ones.

To make clear that S2, which has become S3 in the revised manuscript, deals with the present issue, we have revised its title as follows:

[Before]

“S3. Absorbance characteristics of the RC-PSP”

[After]

“S3. How to determine the DBR layer numbers in the RC-PSP”

We have also added a sentence in the figure caption of S3 to clarify the practical implications of the three different absorbances as follows:

[Added]

“Note that the peak, integrated, and weighted absorbances represent the figure-of-merits appropriate for laser-like, broad-emission-bandwidth, and LED-like excitation sources, respectively.”

► Comment 3

The extinction coefficient of the CQDs would definitely effect the result, especially the blue photon suppression. How would the results differ if perovskites or 2d nano platelets would be utilized rather than conventional QDs.

◀ Our Response

Thanks for the interesting suggestion! As we commented in the manuscript, the RC-PSP is a structurally engineered phosphor, thus ‘not bound to any specific phosphor material.’ In this context, perovskites or 2D nanoplatelets—or anything else—can be also employed as long as they can be physically incorporated into the RC structure, i.e., as long as they can be cast into a thin film of an appropriate thickness ($d = \lambda/2n$) with flat smooth surfaces. If the resultant film has a higher/lower extinction coefficient than CQDs, then a less/greater number of DBR layers would be required in order to obtain similar performance. The optimum RC structure for the corresponding phosphor material can be obtained through thorough absorbance calculations as done for CQDs in the present study.

▶ Comment 4

What is the not-normalized spectra look like? The color coordinate of the white light emission would be better if presented.

◀ Our Response

As pointed out in the manuscript, we have identified full-colour display as the main application sector of the RC-PSP, although white light generation is also possible. For the generation of white colour with good colour coordinates—presumably close to (0.333, 0.333), the three primary colours (R/G/B) should be mixed delicately and correctly in terms of both wavelength and intensity, which would be a formidable task that requires substantial time and efforts.

On the contrary, the full-colour display only requires independent three primary colour sources with high colour purity. All that matters here is the centre wavelengths (for R/G/B colours) and associated spectral linewidths; relative R/G/B intensities required to render a certain colour can be attained by controlling individual colour sources independently. In other words, there is no need to care about the relative intensities of R/G/B! In fact, when the primary colour coordinates (and the corresponding chromaticity colour triangle) of a given display

system are evaluated, three emission spectra from the R/G/B sources are intensity-normalized beforehand, as done in the present study.

To clarify that efficient white light generation could be another important application area for our RC-PSP as the reviewer pointed out, we have revised our manuscript as shown below:

[Before]

“The most obvious and impactful application sector for RC-PSP is the development of full-colour displays based on hybrid micro-LED arrays, in which independently addressable red-green-blue (RGB) pixels can be prepared by combining blue LEDs with red and green phosphors³⁰, as depicted in Fig. 3a.”

[After]

“Although efficient white light generation is certainly a possibility, the most obvious and impactful application sector for RC-PSP should be the development of full-colour displays based on hybrid micro-LED arrays, in which independently addressable red-green-blue (RGB) pixels can be prepared by combining blue LEDs with red and green phosphors³⁰, as depicted in Fig. 3a.”

RESPONSE TO REVIEWER #3

► Comment 0

The paper introduces a novel approach for structurally engineered phosphors, utilizing a one-dimensional (1D) resonant cavity (RC) as a platform. Interestingly, the same author has previously explored engineered phosphors using a photonic crystal (PhC), as mentioned in references 17-19. What is notable is that while the author consistently compares the new results with a Reference Phosphor (Ref-PSP), there is no comparison to their previous work. This raises the question of why the author chose not to compare the new results with those obtained using the photonic crystal.

In reference 17(Adv. Mater. 30, 1703506 (2018)), the author asserts that "This observation indicates that the PhC phosphor is a viable technology for next-generation white LEDs and their applications." However, in the current paper, the author proposes a new platform for structurally engineered phosphors based on a resonant cavity. Once again, the comparison is made with the Reference Phosphor (Ref-PSP) rather than the previously investigated photonic crystal. Does this imply that the statement made in reference 17 is no longer valid and that the new resonant cavity platform (RC) is superior?

It should be noted that the current work falls short of meeting the rigorous editorial criteria for broad impact and significance at Nature Communications. However, with the necessary revisions and corrections (pdf attached), it could still be of interest to a more specialized community.

◀ Our Response

As the reviewer pointed out, we did not compare the RC phosphor with the PhC phosphor, which is the original version of the structurally engineered phosphor we developed. It is simply because the RC phosphor is far more superior than the PhC phosphor in both structure and performance. Details are thoroughly explained below. We hope that our replies are satisfactory in the reviewer's high standards.

► Comment 1

In the paper, the author utilized a different approach by using a resonant cavity, to structurally engineer phosphors. The primary question is: What distinguishes this approach in terms of performance compared to the previously cited work (Ref19)?

Interestingly, both this paper and Ref19 compared same reference sample (Ref-PSP). It raises the question of why a performance comparison between this paper and Ref19 was not conducted. A novel study should ideally showcase advancements over prior research

Is the reference sample used (Ref-PSP) is a standard commercial sample, as mentioned in line 82-83 (ref 30)?

◀ Our Response

Let me outline the brief history of how the structurally engineered phosphor project has evolved in our lab. As a group working on photonic crystals for a long time, we got an idea of how to improve phosphor performance via PhC band-edge modes, which marks the beginning of the structurally engineered phosphor research. Since then, we have been upgrading the PhC phosphor continuously and also seeking for its possible applications, as witnessed by a series of publications (Ref. 16–19). Quite lately, we realized that RC can also play a similar role for phosphor. Further, RC is more attractive than PhC because RC can be fabricated layer-by-layer, which greatly facilitates the realization of the platform and thus strongly appeal for real applications. In good contrast, PhC requires a sophisticated lateral submicron-patterning process, making its practical applications quite challenging. To make a long story short, the RC-PSP is a totally different platform from the PhC phosphor so that we did not and still do not think the direct comparisons between them are really necessary in the present manuscript that deals with the RC-PSP.

As for their performances, it turns out that the RC-PSP performs far better than the PhC phosphor. For example, the RC-PSP used specifically in the present study offers a high absorbance $A \approx 0.85$ —which can be improved even further by increasing cavity Q, whereas the PhC phosphor described in our latest research only exhibits $A \approx 0.5$. All the other performance results, both simulations and experiments, also confirmed the superiority of the RC-PSP. All these can be easily confirmed by comparing the results described in the present manuscript with those of Reference 19.

To clarify the superiority of RC-PSP over PhC phosphor, we have added a paragraph in the revised manuscript as follows:

[Added]

“Furthermore, the RC-PSP is superior to the PhC phosphor in terms of both structure and performance. The RC-PSP can be constructed by vertically stacking planar layers one after another, which is far easier than the sophisticated lateral submicron-patterning process required for fabricating the PhC phosphor. As for the performance, the absorption of excitation photons is greater in the RC-PSP than in the PhC phosphor as revealed hereafter.”

Regarding the reference, on the other hand, we believe that the fair form of the reference phosphor for any structurally engineered phosphor should be the one without any structure (or a simple thin film), but prepared in a similar manner (spin-coating) and containing the same amount of CQDs. This principle was also applied to our previous researches on the PhC phosphors. The reference phosphor for the present study is the cavity layer itself, as described in the manuscript, which differs from that of the previous PhC phosphor—in terms of the CQD amount (or the CQD film thickness). Obviously, it also differs from the one shown in Fig. 1a (i.e., CQDs dispersed in a thick polymer film, which is used in the conventional display colour pixels)—in terms of the structural configuration.

To elucidate the Ref-PSP structure used in the present study more clearly, the manuscript has been revised as follows:

[Before]

“Conversely, the reference phosphor (Ref-PSP), which corresponds to the cavity layer itself—including the two SiO₂ wing layers, exhibits a fairly flat spectral profile with no resonance feature, and the absorbance at $\lambda_0 = 450$ nm is only $A \approx 0.03$ —Fig. 2a(i).”

[After]

“Conversely, the reference phosphor (Ref-PSP), which corresponds to the cavity layer itself—a 40-nm-thick CQD layer cladded by the two SiO₂ wing layers on both sides, exhibits a fairly flat spectral profile with no resonance feature, and the absorbance at $\lambda_0 = 450$ nm is only $A \approx 0.03$ —Fig. 2a(i).”

► Comment 2

In lines 111-113 - compared to the conventional phosphor structure shown in Fig. 1a, the RC-PSP structure is only $\sim 1/10$ of its physical thickness and contains only $\sim 1/100$ of the net CQD amount.

“Where is the reference paper with the conventional phosphor structure that the author is talking about?”

◀ Our Response

Please accept our apology for omitting a proper reference. Our claims in the sentence are based on Ref. 32 (also confirmed by private conversations with the authors of the reference). The manuscript has been revised as follows:

[Before]

“It should be noted that, compared to the conventional phosphor structure shown in Fig. 1a, the RC-PSP structure is only $\sim 1/10$ of its physical thickness and contains only $\sim 1/100$ of the net CQD amount.”

[After]

“It should be noted that, compared to the conventional phosphor structure shown in Fig. 1a, the RC-PSP structure is only $\sim 1/10$ of its physical thickness and contains only $\sim 1/100$ of the net CQD amount³².”

▶ Comment 3

Lines 118-119 - The extinction coefficient of the CQD film determined from independent spectroscopic ellipsometry measurements was used in the calculations.

Is it possible to include a reference or provide the measurements in the supplementary section?

◀ Our Response

We agree that the information on the extinction coefficients (or the entire dispersions of the complex refractive index) of the CQD films should be helpful for readers to understand our work better. To comply with the reviewer’s suggestion, we have added a section in Supplementary Information to provide the n & k dispersions of the both red and green CQD films used in our experiments.

[Added]

S2. Complex refractive indices of the CQD films

Fig. S2 | Dispersion relations of n and k of the CQD films used in the experiments. n and k measured as functions of wavelength by spectroscopic ellipsometry technique are presented for **a, the red CQD film and **b**, the green CQD film.**

► Comment 4

Lines 119-123 - The peak absorbance and resonance linewidth for the RC-PSP were closely coupled and dependent on DRB layer numbers. Therefore, the appropriate RC structure that best fits given excitation requirements must be determined. This study chose the DBR layer numbers as $N_{\text{ex}} = 5$ (or 2.5 pairs) for 122 exDBR and $N_{\text{em}} = 13$ (or 6.5 pairs) for emDBR.

What criterion was used to select this specific number of pairs? Was an LED used as the excitation source? How would the device performance be affected if a laser diode was chosen instead, as mentioned here?

◀ Our Response

The question on the DBR layer numbers was also asked by Reviewer #2 in his/her Comment 2. Once again, Supplementary Information S2 explains how the DBR layer numbers used in the present study are determined. To repeat our answer again, the combination we chose ($N_{\text{emDBR}} + N_{\text{exDBR}} = 5 + 13$) offers high values for all the three figure-of-merits with a relatively small number of the total DBR layers: peak absorbance (suitable for a laser-like excitation source), integrated absorbance (suitable for an excitation source with a constant intensity

profile), and weighted absorbance (suitable for a LED-like excitation source). Therefore, our choice is a compromised solution to make our RC-PSP adaptable broadly to various kinds of excitation sources from LD-like to LED-like ones.

To make clear that S2, which has become S3 in the revised manuscript, deals with the present issue, we have revised its title as follows:

[Before]

“S3. Absorbance characteristics of the RC-PSP”

[After]

“S3. How to determine the DBR layer numbers in the RC-PSP”

We have also added a sentence to the figure caption of S3 to clarify the practical implications of the three different absorbances as follows:

[Added]

“Note that the peak, integrated, and weighted absorbances may represent the figure-of-merits appropriate for LD-like, broad-emission-bandwidth, and LED-like excitation sources, respectively.”

Regarding the excitation source used in our experiments, which is schematically illustrated in the inset of Fig. 2b (i), it is neither LED nor LD, but somewhere between the two. It is LED-

like because it has a broad emission bandwidth (~20 nm), and at the same time it is also LD-like because it is collimated.

Lastly, if LD is used for excitation, the RC resonance can be intentionally designed to be even sharper to obtain higher absorption of excitation photons. However, this condition makes it a formidable task to precisely tune the LD wavelength to the RC resonance peak. We have already made a short comment on this in Ref. 34.

► Comment 5

Lines 177-180 (aRC res. broadening) - However, linewidth broadening is not necessarily a disadvantage because phosphors are typically excited by a light source with a broad bandwidth, such as an LED, for which an RC with a comparable resonance linewidth is preferable to maximise the use of excitation photons.

However, in reference 34 that you cited, the use of a laser diode as the excitation source was mentioned. In this case, how would the broadening effect impact the performance of your device? Additionally, why is this point significant when you previously stated that phosphors are typically excited by a light source with a broad bandwidth? You also mentioned in lines 103-104 the use of the laser diode.

◄ Our Response

In principle, a LD-like excitation source, characterized by a sharp emission linewidth (typically less than 1 nm), should be ideal for our RC-PSP. RC can then be designed such that its resonance matches exactly with the LD emission in terms of both wavelength and linewidth to induce near 100% absorption of excitation photons. In practice, however, the exact wavelength tuning becomes extremely difficult when the both linewidths (LD emission and RC resonance) are narrow. Meanwhile, most of the phosphor excitation sources employed in real and practical devices are LEDs as mentioned in our manuscript, presumably because of economic reasons. In the view point of operating the RC-PSP, a LED-like excitation source makes the wavelength tuning far easier because its linewidth is much broader (> 10 nm). It is in this context that we said in our manuscript that a broadened resonance linewidth is not necessarily a disadvantage. Therefore, characteristics of a given excitation source should/could

be properly reflected in the RC design, for example, by choosing a suitable figure-of-merit among the maximum/integrated/weighted absorbances.

► Comment 6

Lines 198-200 (camera image) – The CQD fluorescence intensities of RC-PSP and α RC-PSP significantly exceeded that of Ref-PSP.

Could you please provide the intensity values for the numbers? It is evident from the image that a comparison between RC-PSP and α RC-PSP with the reference Ref-PSP shows clear differences. However, when comparing RC-PSP with α RC-PSP, it is not as apparent.

◀ Our Response

The intensity information that the reviewer inquired can be found in Fig. 2b, where detailed PLE data are shown in 3D format.

Considering our manuscript in the reviewer's (or readers') point of view, however, the authors have realized that the quantitative fluorescence intensity comparisons among Ref-PSP, RC-PSP, and α RC-PSP are difficult to make from both the PLE data (Fig. 2b) and the photograph images (Fig. 2c). In order to facilitate such quantitative comparisons and also to

comply with the reviewer’s request, we have added a table along with a sentence to explain about it in the revised manuscript.

[Added]

“The CQD fluorescence intensities of RC-PSP and α RC-PSP significantly exceeded that of Ref-PSP. For quantitative comparisons, both the peak and integrated fluorescence intensities for Ref-PSP, RC-PSP, and α RC-PSP are extracted from the PLE data in Fig. 2b and summarized in Table 1.”

Table 1. Peak and integrated fluorescence intensities for Ref-PSP, RC-PSP, and α RC-PSP

	Ref-PSP	RC-PSP	α RC-PSP
Peak Intensity	190	6,070	7,220
Integrated Intensity	6,530	191,650	209,720

*The numbers are deduced from the PLE data in Fig. 2b.

► Comment 7

Lines 216-218 - Hence, the high-index DBR material TiO₂ was replaced with Ta₂O₅ (nTaO₅ ≈ 2.08), while the numbers of DBR layers were increased to N_{ex} = 7 (3.5 pairs) and N_{em} = 31 (15.5 pairs) to compensate for the lowered index contrast.

Will the number of DBR pairs significantly affect the device's cost in comparison to the initial RC-PSP design (6.5 to 15.5 pairs), considering its intended use in real-world applications?

◀ Our Response

As clarified in the Methods section in the manuscript, the DBRs are vacuum-deposited using e-gun evaporator. Due to the nature of vacuum process, it is charged by runs, not by the number of layers. In other words, an increase in the DBR layer numbers would not affect the cost at all.

► Comment 8

Lines 229-231 - As shown in Figs. 3b and 3c, the FEFs are as high as ~6.8 and ~5.9 for the red and green RC-PSPs, respectively. Compared with the results shown in Fig. 2e, which were

obtained with $\lambda_{ex} \approx 2$ nm, the maximum FEFs are much lower, while the resonance linewidths become larger ($\lambda_0 \approx 20$ nm).

The performance significantly decreased when an LED was used as the excitation source. How can this issue be addressed and resolved?

◀ Our Response

This issue was already discussed in our response to Comment 5. In short, the narrower the emission linewidth of excitation source and the resonant linewidth of RC, the better the RC-PSP performance. This argument is consistent with our experimental results performed with two different excitation linewidths: 2 nm and 20 nm. This is an intrinsic effect, implying that there is nothing we can do about it, but we have to live with it. Nonetheless, please note that the fluorescence enhancement factors of ~ 6.8 and ~ 5.9 are already quite impressive numbers! In addition, we would like to remind the reviewer that our RC-PSP structure is a compromised design for both LD-like and LED-like excitation sources, as explained in our Response to Comment 4. Therefore, we could redesign our RC for optimised or best performance when the excitation source is fixed and its characteristics are given.

▶ Comment 9

Line 288 – Fabrication of the RC-PSP

Please detail more the fabrication process: DBR thickness, growth rate, etc. The information should be available or a reference point it.

◀ Our Response

We believe that the RC-PSP fabrication details are already well described in the Methods section, which is copied below.

Fabrication of the RC-PSP

Device fabrication involved three simple steps: two vacuum depositions and one spin-coating in between. First, the $\text{TiO}_2/\text{SiO}_2$ (or $\text{Ta}_2\text{O}_5/\text{SiO}_2$) emDBR and the first SiO_2 wing layer were deposited on a fused quartz substrate using an e-gun evaporator at an elevated temperature of $T = 150^\circ\text{C}$. During the deposition of the high-index layers, an O_2 environment was provided to

fully oxidise the metal ions. Subsequently, CdSe-ZnS core-shell CQDs (CZO-620H and CZO-530H, Zeus) dispersed in a cyclohexane solution were spin-coated on top of the first SiO₂ wing layer. The CQD concentration (2 wt. %) and spin speed (3000 rpm) were carefully selected to obtain the desired CQD film thickness (40 nm). Finally, another round of vacuum deposition of the second SiO₂ wing layer and exDBR was performed to complete the device fabrication. The first and last layers of both emDBR and exDBR was a high-index layer (TiO₂ or Ta₂O₅). For α RC-PSP, exDBR and α DBR were simultaneously deposited in a single run.

The information on the DBR layer thicknesses can be found in the main body of the manuscript, which is also copied below.

The two DBRs consist of alternating $\lambda/4$ -thick dielectric layers of TiO₂ ($d_{\text{TiO}_2} \approx 49$ nm, $n_{\text{TiO}_2} \approx 2.32$) and SiO₂ ($d_{\text{SiO}_2} \approx 77$ nm, $n_{\text{SiO}_2} \approx 1.47$), starting and ending with the TiO₂ layers for the higher index contrasts to the environmental materials, air and silica.

The only information that the reviewer specified but we cannot provide is the growth rate (or deposition rate) of the DBR layers because it is the proprietary information of the company from which we outsourced the vacuum depositions. Nonetheless, a typical deposition rate for dielectric films in e-gun evaporator is a few Å/s. We are sure that the deposition rates the company used for our samples is also within the range. If the reviewer still feels that further details should be provided, please be specific and we will happily answer.

► Last Comment

Given the concern over the toxicity of Cd, there has been a growing focus on developing environmentally friendly QLEDs that are free from heavy metals. Why should you continue to pursue this direction?

◀ Our Response

We are well aware that the use of Cd-based CQDs are now tightly controlled due to ever-growing environmental issues and even strictly forbidden in commercial products. However, the Cd-free CQDs, such as InP-based ones, are not only expensive but also difficult to procure, whereas Cd-based conventional CQDs are readily available commercially at an affordable

price. Naturally, university labs have been still using Cd-based CQDs for R&D purposes and in small volumes. Please be reminded that the concept of the structurally engineered phosphors is not bound to Cd-based CQDs but applicable to any colour-converting materials. It is only because of the availability and affordability that we have stuck to Cd-based CQDs.

===== End of Report =====

Final revisions

12th September

REVIEWERS' COMMENTS

Reviewer #1 (Remarks to the Author):

Accept

Reviewer #2 (Remarks to the Author):

The authors have addressed all queries scientifically during the revision, therefore from my perspective the paper can now be accepted for publication.